# *Q*-dependent collective relaxation dynamics of glass-forming liquid Ca$_{0.4}$K$_{0.6}$(NO$_3$)$_{1.4}$ investigated by wide-angle neutron spin-echo

Peng Luo [1], Yanqin Zhai [1,2], Peter Falus [3], Victoria García Sakai [4], Monika Hartl[5], Maiko Kofu [6], Kenji Nakajima [6], Antonio Faraone [7✉] & Y Z [1,2,8✉]

The relaxation behavior of glass formers exhibits spatial heterogeneity and dramatically changes upon cooling towards the glass transition. However, the underlying mechanisms of the dynamics at different microscopic length scales are not fully understood. Employing the recently developed wide-angle neutron spin-echo spectroscopy technique, we measured the *Q*-dependent coherent intermediate scattering function of a prototypical ionic glass former Ca$_{0.4}$K$_{0.6}$(NO$_3$)$_{1.4}$, in the highly viscous liquid state. In contrast to the structure modulated dynamics for $Q < 2.4$ Å$^{-1}$, i.e., at and below the structure factor main peak, for $Q > 2.4$ Å$^{-1}$, beyond the first minimum above the structure factor main peak, the stretching exponent exhibits no temperature dependence and concomitantly the relaxation time shows smaller deviations from Arrhenius behavior. This finding indicates a change in the dominant relaxation mechanisms around a characteristic length of $2\pi/(2.4$ Å$^{-1}) \approx 2.6$ Å, below which the relaxation process exhibits a temperature independent distribution and more Arrhenius-like behavior.

[1] Beckman Institute for Advanced Science and Technology, University of Illinois at Urbana-Champaign, Urbana, IL 61801, USA. [2] Department of Nuclear, Plasma, and Radiological Engineering, University of Illinois at Urbana-Champaign, Urbana, IL 61801, USA. [3] Institut Laue-Langevin (ILL), 38042 Grenoble, France. [4] ISIS Neutron and Muon Facility, Rutherford Appleton Laboratory, Science & Technology Facilities Council, Didcot OX11 0QX, UK. [5] European Spallation Source, SE-221 00 Lund, Sweden. [6] J-PARC Center, Japan Atomic Energy Agency, Tokai, Ibaraki 319-1195, Japan. [7] NIST Center for Neutron Research, National Institute of Standards and Technology, 100 Bureau Drive, Gaithersburg, MD 20899-1070, USA. [8] Department of Electrical and Computer Engineering, University of Illinois at Urbana-Champaign, Urbana, IL 61801, USA. ✉email: antonio.faraone@nist.gov; zhyang@illinois.edu

Glassy materials are ubiquitous in nature and important for various technologies, however, our understanding of the nature of a glass and the physical mechanisms underlying the process of glass formation remains elusive even after decades of intensive study. What is behind the core problem, is the fact that upon cooling liquids towards the glass transition, the relaxation time (or viscosity) exhibits extraordinary growth but the microscopic structure undergoes only minute changes[1]. Starting from the microscopic equations of motion of a correlated liquid, the mode-coupling theory[2–4] predicts a two-step relaxation process in a moderately viscous liquid state, introduces the ergodicity parameter, the plateau level separating the two dynamic processes as a defining parameter of the glass transition, and makes quantitative predictions on the dynamical behavior approaching the glass transition at the critical temperature $T_c$. However, it cannot account properly for some complex dynamical phenomena in the high-viscosity regime approaching the experimentally observed glass transition temperature. It has been proposed that the relaxation dynamics of supercooled liquids are spatially heterogeneous, where the individual relaxing units have distinct, site-specific relaxation times, with a characteristic length scale that grows with decreasing temperature[5–12]. Previous studies have suggested that the length scale associated with dynamic heterogeneity of supercooled liquids is in the region of 1–3 nm upon approaching the glass transition temperature[13–16]. To establish a detailed atomic-level description of the complex relaxation dynamics of glass-forming liquids, a *direct* measurement of the relaxation processes at different microscopic length scales beyond the hydrodynamic regime[17] is required.

Another example indicating the imperative importance of the length scale dependence study is the reported crossover from microscopic dynamics dominated by the slow α-relaxation, to the Johari-Goldstein (JG) β-relaxation at a certain length scale as the temperature is decreased in the intermediate supercooled liquid region. Via neutron spin-echo (NSE)[18–20] and nuclear resonance X-ray scattering (NRXS)[21–23], this slow α and JG β crossover which occurrs in the time scale of ≈10 ns to ≈$10^3$ ns has been directly observed, and its occurrence has been found to be wavevector (Q)-dependent. This crossover occurs in the Q-range of the first minimum above the primary peak of the structure factor $S(Q)$, but around the primary peak it is either not observed[21,22] or is too weak to be appreciated[23], due to the local length scale of the JG β-process. The technique of NSE, which encodes the dynamic signal in the spin of the neutrons and measures the Q-dependent intermediate scattering function (ISF) directly in time domain, offers the highest energy resolution among all neutron spectroscopy techniques and thus has been extensively employed for studying the relaxation dynamics of a wide variety of materials on the pico- to nanosecond time scale[19,24–36]. Traditional NSE instruments only measure one Q at a time, thus limiting the number of accessible Q values in a given experiment. Thus, the question still remains as to what the underlying mechanism of the relaxations is at different microscopic length scales.

In this work, we employed the recently developed and commissioned Wide-Angle Spin-echo (WASP) instrument[37] at the Institut Laue-Langevin (ILL), which employs anti-Helmholtz coils to allow the simultaneous measurement of the coherent ISF over a broad Q-range with unprecedented high data rate and accuracy[37], to scrutinize the collective relaxation dynamics of $Ca_{0.4}K_{0.6}(NO_3)_{1.4}$ [CKN, onset glass transition temperature $T_g \approx$ 336 K as measured using differential scanning calorimetry (Supplementary Fig. 1) and liquidus temperature $T_L \approx 483$ K (Ref. [31])], in the highly viscous liquid states, over the full range of the structure factor main peak and the following valley, specifically between $Q = 1.08$ Å$^{-1}$ and $Q = 2.82$ Å$^{-1}$. The measured

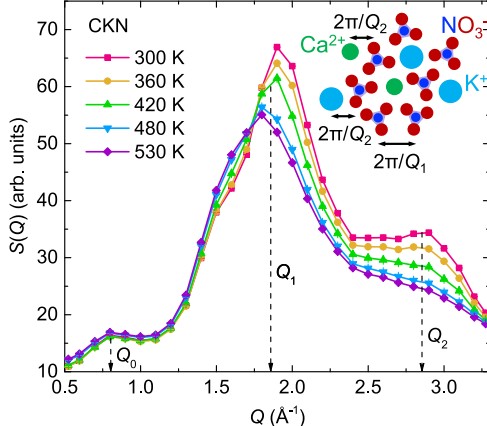

**Fig. 1 Static structure factor of CKN obtained from neutron scattering.** The measurements were performed at five different temperatures from the glass (300 K) up to the low-viscosity liquid (530 K) states. The inset is an illustration of CKN structure with indicated correlation lengths; the triangles represent nitrate ions $NO_3^-$ with oxygen atoms as red circles and nitrogen atoms as blue circles, the sky-blue and green circles are potassium ions K$^+$ and calcium ions Ca$^{2+}$, respectively; the arrow lines represent the length scales corresponding to the primary peak around $Q_1 \approx 1.86$ Å$^{-1}$ originating mainly from the correlation of neighboring nitrate ions, and the broad peak at $Q_2 \approx 2.85$ Å$^{-1}$ manifesting mainly the ionic oxygen-cation correlations. The illustration for the length scale corresponding to the pre-peak at $Q_0 = 0.8$ Å$^{-1}$ arising from the intermediate-range aggregates of nitrate ions is not shown. Error bars represent one standard error and are smaller than the symbol size. Source data are provided as a Source Data file.

coherent ISF is comprised of both the self and distinct components, and therefore reflects motions that are not necessarily cooperative[38,39]. CKN represents a prototypical fragile glass-former in which coherent neutron scattering dominates, therefore its relaxation dynamics has been extensively studied using neutron scattering[29,30,40–43]. However, due to the limited range of the Q-t-T space and data accuracy probed by previous measurements, the questions such as whether and how the relaxation dynamics are modified at different length scales, still remain unclear[40,41]. The data reported here with high accuracy and full mapping of a wide range of time, temperature, and length scale, reveal interesting phenomena for the microscopic relaxation dynamics and shed new light on the microscopic dynamics evolution of glass-forming liquids upon cooling towards the glass transition.

## Results

**Static structure factor of CKN.** Figure 1 shows the static structure factor $S(Q)$ of CKN at various temperatures between the glassy (300 K) and the low-viscosity liquid (530 K) states, obtained by integration of the dynamic structure factor in the energy range of [−5, 0] meV measured on the cold-neutron disk-chopper spectrometer AMATERAS[44,45] at J-PARC Japan, using a neutron wavelength of 3.26 Å. In agreement with the previous neutron and X-ray diffraction studies[42,46,47], three broad peaks are observed in $S(Q)$ of CKN: A pre-peak at $Q_0 = 0.8$ Å$^{-1}$ indicative of intermediate-range ordering, the primary peak at $Q_1 \approx 1.8$ Å$^{-1}$ to 1.9 Å$^{-1}$ depending on temperature, and another broad peak at $Q_2 \approx 2.85$ Å$^{-1}$. Since nitrogen and oxygen have much larger neutron scattering cross sections (see Supplementary Table 1) as well as higher concentrations in CKN than calcium and potassium, the neutron scattering signals in CKN are mainly from nitrogen and oxygen. The previous study on CKN by Tengroth et al.[47] employing neutron diffraction and reverse Monte Carlo modeling suggests that, the pre-peak at 0.8 Å$^{-1}$

could arise from the chainlike aggregates formed by orientationally correlated neighboring nitrate ions such that the oxygen atoms are facing each other; the primary peak is to a large extent the structure of the nitrate ions, and the third peak at 2.85 Å$^{-1}$ manifests mainly the ionic oxygen-cation correlations.

**Intermediate scattering functions**. Wide-angle NSE measurements for CKN were carried out using the WASP spectrometer at various temperatures between 383 K and 519 K. Additional NSE measurements extending the $Q$-range to the pre-peak of $S(Q)$ at $Q = 0.8$ Å$^{-1}$ were performed using the NGA-NSE spectrometer[48] at the National Institute of Standards and Technology (NIST) Center for Neutron Research (NCNR). To access shorter times to complement the NSE data, Fourier deconvoluted time-of-flight spectra measured on the AMATERAS neutron spectrometer[44,45] at the J-PARC facility were compared with the ISFs collected on the WASP spectrometer. Nuclear neutron scattering is composed of a coherent component yielding information on the relative position and motion of couples of atoms, and an incoherent component related to the single particle dynamics of the atoms[24,49]. Polarized diffraction that can separate coherent and incoherent contributions to the scattering signal was performed for CKN on the NGA-NSE spectrometer at the NCNR. The data confirmed that coherent scattering overwhelmingly dominates in the NSE measurements of CKN over the studied $Q$-range (see Supplementary Fig. 2 and related calculation in the Supplementary Information). More details about the experimental methods are included in the Methods part.

Figure 2 shows representative ISFs, $I(Q,t)/I(Q,0)$, of CKN at $T = 444$ K for various $Q$'s [Fig. 2a], and at the structure factor maximum of $Q = 1.86$ Å$^{-1}$ at various temperatures [Fig. 2b]. In comparison to previous data over the same time window[40,41], our data collected on the WASP spectrometer show much improved

statistics and signal-to-noise ratio and reveal unambiguously the existence of a two-step relaxation. As can be seen from Fig. 2a, both the shape and amplitude of the slow α-process are highly dependent on $Q$, indicating a length scale relevant relaxation dynamics. Figure 2b indicates that even in the liquid state ($T > T_L$), the relaxation of CKN exhibits a well-defined two-step process. The ISFs collected on the NGA-NSE spectrometer are reported in Supplementary Fig. 3.

**Fitting model for intermediate scattering functions**. In previous pioneering studies of CKN by Mezei[40,41], it was shown that fits using a single stretched exponential function to account for a combination of a fast process and the slow α-process, were invalid. Therefore, for the data from this work we employ a fitting procedure[28,50,51] which combines two exponential functions, a simple exponential that describes the initial fast process and a stretched exponential function that describes the slow α-process:

$$\frac{I(Q,t)}{I(Q,0)} = \left[1 - f\right]\exp\left[-\left(\frac{t}{\tau_{\text{fast}}}\right)\right] + f\exp\left[-\left(\frac{t}{\tau_{\text{slow}}}\right)^{\beta}\right] \quad (1)$$

where $f$ is the amplitude of the slow α-relaxation (the effective Debye–Waller factor), $\tau_{\text{fast}}$ and $\tau_{\text{slow}}$ are the relaxation times of the fast and the slow α-process respectively, and $\beta$ the shape exponent of the α-relaxation. The fittings were performed leaving all the four parameters free. Attempts to fit the data with only a stretched exponential function was not successful even for the highest temperature at 519 K.

**$Q$-dependence of the relaxation dynamics**. The fitted $f$, $\tau_{\text{slow}}$, and $\beta$ are plotted against $Q$ in Fig. 3, which indicates that all parameters describing the density correlations are significantly $Q$-dependent and exhibit pronounced characteristic oscillations in phase with $S(Q)$. For the lowest temperatures of 383 K and 394 K, and at $Q = 1.08$ Å$^{-1}$ and $T = 402$ K, the α-relaxation is outside the accessible time window of the measurement, therefore, the values of $\tau_{\text{slow}}$ and $\beta$ were excluded. However, since the initial plateau of the α-process is observed directly in these cases, the determination of $f$ is possible.

The α-relaxation time $\tau_{\text{slow}}$ follows an $S(Q)/Q^2$ dependence above $Q = 1.6$ Å$^{-1}$ at all temperatures, see Fig. 3a, and Supplementary Fig. 4 of the normalized data. This is in agreement with the previous observations in other glasses[32,52], and suggests a local order modified diffusive-like process at the length scales of $Q > 1.6$ Å$^{-1}$. Such a behavior is similar to the so-called de Gennes narrowing[53], an effect predicted from the *short* time expansion of the time correlation function and therefore not expected for the slow processes; more theorical efforts would be needed for a rigorous explanation of these observations on the time scale of the α-relaxation. Below 1.6 Å$^{-1}$, $\tau_{\text{slow}}$ is larger than that predicted by the $S(Q)/Q^2$ dependence, an occurrence likely related to the additional influence of the intermediate range ordering as indicated by the pre-peak of $S(Q)$ around $Q = 0.8$ Å$^{-1}$ (Fig. 1). It has been previously suggested by Mezei et el.[54] that around the pre-peak the flow of ionic groups occurs in a string-like fashion, consistent with the observation of chain-like aggregates of nitrate ions by Tengroth et al.[47].

Figure 3b shows that at the studied temperatures $\beta$ also varies in phase with $S(Q)$, between ≈0.45 and ≈0.75, being largest around the maximum of $S(Q)$. Such a behavior has also been observed in metallic supercooled liquids close to $T_g$ by X-ray photon correlation spectroscopy[55], suggesting a universal dynamical property across a large timescale range covering 16 orders of magnitude (from ≈1 ps to >10$^3$ s). The $Q$-dependence of $\beta$ was also observed in *ortho*-terphenyl[32], isopropanol[56], hard-sphere system[57] and binary soft-sphere mixture[58], indicating again the

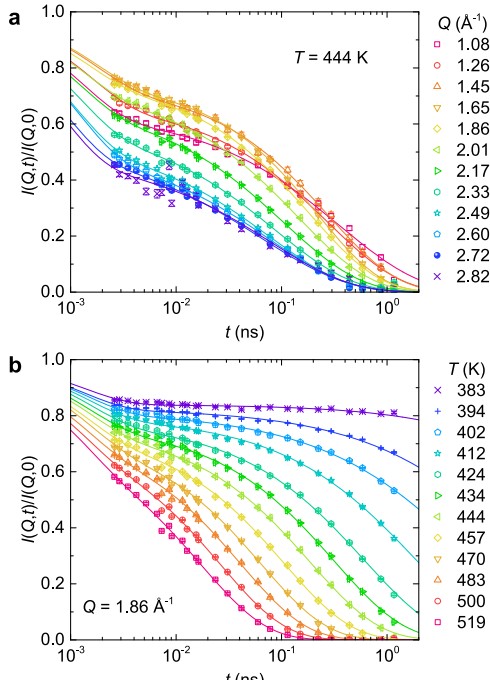

**Fig. 2 Representative intermediate scattering functions of CKN.**
**a** $T = 444$ K at various $Q$'s between 1.08 Å$^{-1}$ and 2.82 Å$^{-1}$, **b** $Q = 1.86$ Å$^{-1}$ at various temperatures between 383 K and 519 K. The solid lines represent fit to Eq. (1). Error bars represent one standard error and are smaller than the symbol size. Source data are provided as a Source Data file.

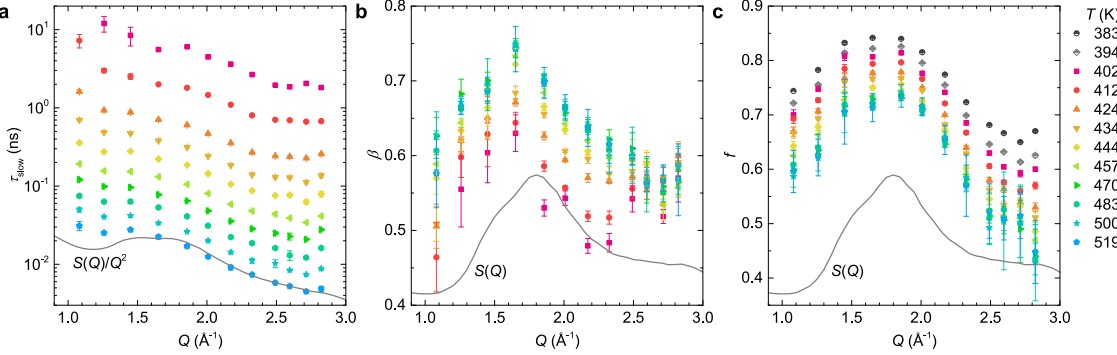

**Fig. 3 Q-dependence of the relaxation dynamics at different temperatures.** (**a**) relaxation time $\tau_{slow}$, (**b**) exponent $\beta$, and (**c**) effective Debye–Waller factor $f$ as a function of $Q$ at various temperatures. The legend in **c** applies to all three panels. The solid line in **a** represents $S(Q)/Q^2$, and those in **b** and **c** are $S(Q)$, the static structure factor measured at $T = 480$ K. Error bars represent one standard error and where not seen they are smaller than the symbol size. Source data are provided as a Source Data file.

important effect of particle-particle correlations on the collective relaxation dynamics. At the length scale specified by the structure factor peak, a narrower distribution of possible relaxation processes and thus larger $\beta$ is observed, reflecting the more pronounced structure correlations of the liquid at this $Q$. However, at $Q$'s outside this length scale, the structural correlations become weaker, leading to a wider distribution of the relaxation processes as manifested by the decreased value of $\beta$. As shown in Fig. 3c, $f$ varies between ≈0.45 and ≈0.85 and shows a systematic variation in phase with $S(Q)$, which is consistent with the previous NSE measurements of Mezei et al.[30].

**T-dependence of the relaxation dynamics.** Figure 4 reports the temperature dependence of the fitting parameters for the data measured on both the WASP spectrometer and the NGA-NSE spectrometer. To improve readability, some of the data for $\beta$ and $\tau_{slow}$ are reported in Supplementary Figs. 5 and 6, and the plot of $\beta$ as a function of temperature is divided into two panels. The left panel in Fig. 4a shows $\beta$ as a function of temperature for $Q \leq 2.33$ Å$^{-1}$, where one can see that at $T > 470$ K $\beta$ is unaffected by temperature change, while it decreases with decreasing temperature at $T < 470$ K; these findings indicate an increased distribution of the relaxation time, in agreement with the enhanced dynamic heterogeneity of the α-relaxation in the highly viscous liquid state[7]. This behavior is fully confirmed by the NSE measurements at NCNR, at $Q = 0.80$ Å$^{-1}$ and $1.75$ Å$^{-1}$ (half-filled triangles). However, in stark contrast, we can see in the right panel of Fig. 4a that, for $Q = 2.49$ Å$^{-1}$ $\beta$ also decreases with decreasing temperature at $T < 470$ K, but at a much slower rate than that at lower $Q$'s; eventually, for $Q \geq 2.60$ Å$^{-1}$, $\beta$ shows no systematic change with temperature. The slope of the linear fit to the temperature dependence of $\beta$ (d$\beta$/d$T$) at $T \leq 470$ K for $Q \leq 2.49$ Å$^{-1}$ and at the entire temperature range for $Q > 2.49$ Å$^{-1}$ is shown in the lower panel of Fig. 4d.

As reported in Fig. 4b we can see that at each $Q$ the effective Debye–Waller factor $f$ shows a linear decrease with increasing temperature until $T = 470$ K, above which it levels off. This observation is confirmed by the NSE measurements at NCNR at $Q = 0.80$ Å$^{-1}$ and $1.75$ Å$^{-1}$ (half-filled triangles), albeit with a small offset to the WASP data. Our data corroborate the previous result of a continuous decrease of $f$ with increasing temperature by Mezei et al.[30] (open diamonds), and further demonstrate that a constant $f$ indeed exists but only at temperatures above 470 K, which was not established from the previous results of CKN[30,40,41].

Figure 4c is the Arrhenius plot of $\tau_{slow}$ fitted with the Vogel–Fulcher–Tammann (VFT) law[59],

$$\tau_{slow} = A\exp\left(\frac{DT_0}{T - T_0}\right) \qquad (2)$$

where $A$, $D$, and $T_0$ are the fitting parameters. In Angell's fragility classification, CKN is a typical fragile glass-former in which the relaxation dynamics deviate from the Arrhenius behavior significantly[59]. In the studied temperature range, $\tau_{slow}$ can be well fitted by the VFT law [Fig. 4c]. The resulting $Q$-dependent $D$ and $T_0$ are plotted in Fig. 4d as red circles, together with dash lines that represent respectively $D = 1.53 \pm 0.15$ and $T_0 = 340.0 \pm 3.5$ K as obtained by VFT fit to the macroscopic shear relaxation time $\langle\tau\rangle = \eta/G_\infty$ in the temperature range of the present study [Supplementary Fig. 7a], using the previously reported shear viscosity $\eta$ (Ref. [60,61]) and limiting high-frequency shear modulus $G_\infty$ (Ref. [62]). We can see that the $D$ and $T_0$ obtained from the macroscopic shear relaxation time agree with the mean value of those at the lower $Q$'s, indicating that the macroscopic relaxation in CKN reflects the average of the processes at the different microscopic length scales. At $Q < 2.4$ Å$^{-1}$, the $Q$-dependence of $D$ follows the shape of the static structure factor $S(Q)$, and that of $T_0$ and d$\beta$/d$T$ follow the shape of $-S(Q)$, i.e., the opposite trend of $S(Q)$. While at $Q > 2.4$ Å$^{-1}$, $D$ shows a rapid increase with increasing $Q$, and $T_0$ and d$\beta$/d$T$ a rapid decrease. Forcing either $T_0$ or $D$ to take constant values like those obtained from the macroscopic shear relaxation time, prevents a satisfactory fit of the data, see Supplementary Fig. 8 for example, confirming that $T_0$ and $D$ are inherently $Q$-dependent.

Note that at $Q > 2.4$ Å$^{-1}$ the exponent $\beta$ shows rather weak or absent temperature dependence (d$\beta$/d$T = 0$), in contrast to that at lower $Q$'s [Fig. 4a]. Therefore, to consider the influence of $\beta$, we calculated the mean relaxation time,

$$\langle\tau_{slow}\rangle = \frac{\tau_{slow}}{\beta}\Gamma\left(\frac{1}{\beta}\right) \qquad (3)$$

being $\Gamma(x)$ the gamma function, and fitted it with the VFT expression (Supplementary Fig. 9). As the black squares in Fig. 4d indicate, the influence of $\beta$ on the evaluation of $D$ and $T_0$ is negligible, the in-phase variation of $D$ with $S(Q)$ and of $T_0$ with $-S(Q)$ at $Q < 2.4$ Å$^{-1}$, as well as their rapid changes at $Q > 2.4$ Å$^{-1}$ are retained for the case of $\langle\tau_{slow}\rangle$. These results, as manifested by the $Q$-dependence of $D$, $T_0$, and d$\beta$/d$T$ in Fig. 4d, indicate a transition of the relaxation behavior of the liquid at a characteristic spatial length of $2\pi/Q \approx 2.6$ Å ($Q = 2.4$ Å$^{-1}$).

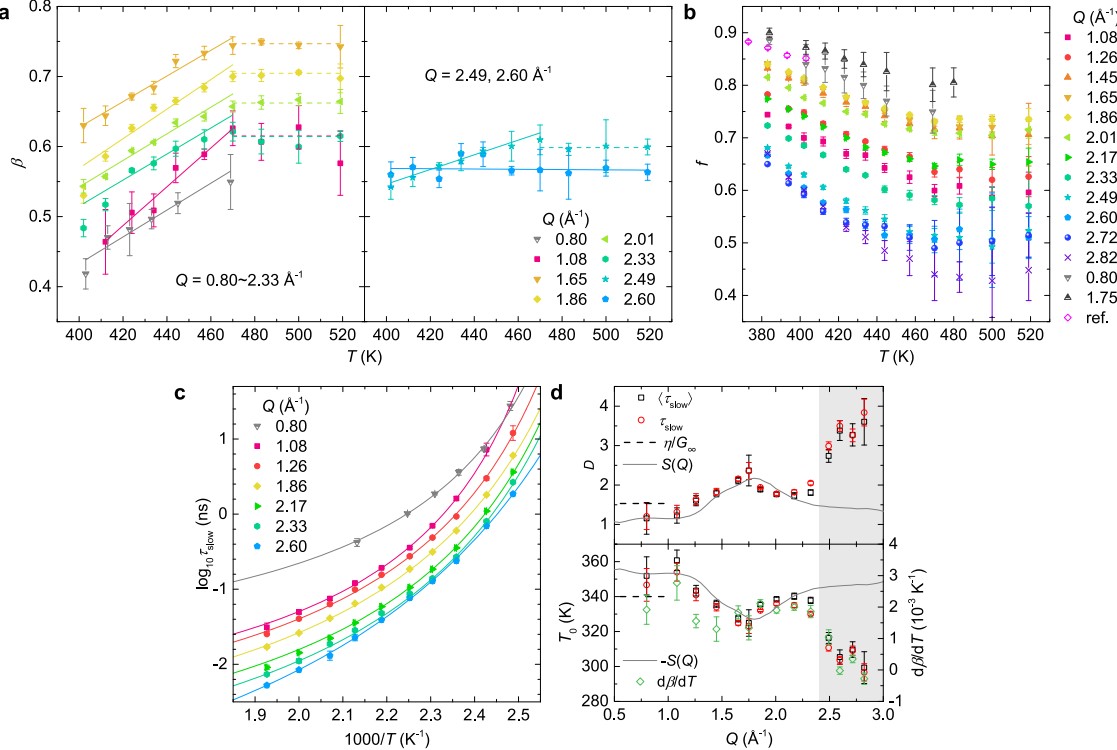

**Fig. 4 Temperature behavior of the relaxation dynamics at different length scales. a** Exponent $\beta$ as a function of temperature (left panel: $Q = 0.80$ Å$^{-1}$ to 2.33 Å$^{-1}$, right panel: $Q = 2.49$ Å$^{-1}$ and 2.60 Å$^{-1}$, solid lines are representative linear fit and dash lines are guide for the eye). **b** Temperature dependence of effective Debye–Waller factor $f$ at various $Q$'s. The open diamonds are adapted from Ref. [30] at $Q = 1.86$ Å$^{-1}$. **c** Logarithmic relaxation time $\tau_{slow}$ as a function of $1000/T$ with VFT fits (Eq. (2)). To make it easier for the reader to discern the curves, the data for only representative $Q$'s are shown in **a** and **c**, the rest of the data can be found in Supplementary Figs. 5 and 6. The half-filled triangles in **a**–**c** represent the results measured on NGA-NSE. **d** $Q$-dependence of $D$ (upper panel) and $T_0$ (lower panel) obtained from the VFT fit to $\tau_{slow}$ (red circles), $\langle\tau_{slow}\rangle$ (black squares), and the macroscopic shear relaxation time $\langle\tau\rangle = \eta/G_\infty$ (dash lines) [see Supplementary Figs. 7a and 9 for the VFT fit to $\langle\tau\rangle$ and $\langle\tau_{slow}\rangle$], diamonds in the lower panel are $Q$-dependence of d$\beta$/d$T$ (right axis). The solid line in the upper panel of **d** is the static structure factor $S(Q)$ measured at $T = 480$ K, and that in the lower panel is $-S(Q)$. Error bars represent one standard error and where not seen they are smaller than the symbol size. Source data are provided as a Source Data file.

## Discussion

Previous mechanical spectroscopy suggested that the JG $\beta$-relaxation, which follows Arrhenius behavior, exists in CKN[63]. However, for all $Q$'s studied here, $\tau_{slow}$ follows a non-Arrhenius VFT behavior in the entire studied time range. Thus, our observation of contrasting behaviors of d$\beta$/d$T$, $D$, and $T_0$ beyond the first minimum above the primary peak of $S(Q)$ (i.e., $Q > 2.4$ Å$^{-1}$) with respect to that around the primary peak, indicate that the modifications of relaxation mechanisms at the local length scales of $2\pi/Q < 2.6$ Å, are not trivial at all even at temperatures well above the α to JG $\beta$ crossover, which was estimated to be around 375 K for CKN by extrapolating the reported data from macroscopic measurements[60–64] [see Supplementary Fig. 7b]. The increase of $D$ and the concomitant decrease of $T_0$ at $Q > 2.4$ Å$^{-1}$ indicate smaller deviations of the relaxation dynamics from Arrhenius behavior, that is, a more Arrhenius-like behavior at the local length scales. This can also be directly appreciated without the VFT fit as in Fig. 5, reporting the Arrhenius plot with the relaxation time normalized to a reference temperature ($T = 402$ K). In agreement with the VFT fitting results, less discrepancies from the Arrhenius behavior (black dash line) are evident at larger $Q$'s (2.49 Å$^{-1}$ and 2.82 Å$^{-1}$), in both cases of $\tau_{slow}$ [Fig. 5a] and $\langle\tau_{slow}\rangle$ [Fig. 5b], reinforcing the conclusion that the dynamics at the local length scales behave more Arrhenius-like than at larger length scales.

Deviations from Arrhenius behavior indicates a temperature dependent activation energy. A smaller deviation from Arrhenius

behavior observed for the higher $Q$'s ($Q > 2.4$ Å$^{-1}$) indicates that the activation energy at the local length scales is less temperature dependent than at larger length scales, i.e., at higher temperatures the activation energy at the local length scales is larger than at larger length scales, in contrast, at lower temperatures it is smaller. This could be associated with the fact that the local environment is relatively more stabilized against temperature change due to the Coulombic interactions, leading to a more Arrhenius-like behavior, as it will be discussed more in details in the following. Moreover, the rather slow decrease or even invariant exponent $\beta$ with decreasing temperature below $T_L$ for $Q > 2.4$ Å$^{-1}$, suggests that the dynamic heterogeneities at the local length scales are unaffected by temperature, in contrast to the slow α-relaxation at $Q < 2.4$ Å$^{-1}$. Analogously, the observation that, $D$ reaches a maximum and, $T_0$ and d$\beta$/d$T$ reaches a minimum around the primary peak of $S(Q)$ [Fig. 4d], indicates that the activation energy for the structural relaxation and the relaxation processes distribution are less temperature dependent. It means that, not only the parameters that describe the relaxation process at a constant temperature (Fig. 3), but also their temperature behavior, could be affected by the inter-particle correlation of the nitrate ions which gives rise to the primary peak of $S(Q)$, further evidencing the strong structure-dynamics correlation at the microscopic length scales.

We note that the information gathered from our results was not available in previous NSE studies, where a $Q$-independent and/or $T$-independent $\beta$ was often assumed a priori owing to the

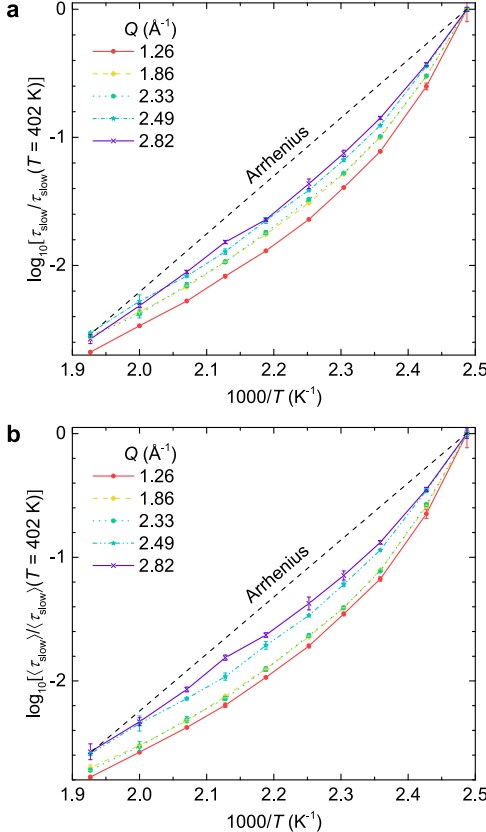

**Fig. 5 Direct comparison of the temperature dependence of the relaxation time at different Q's.** (**a**) Relaxation time $\tau_{slow}$ and (**b**) average relaxation time $\langle\tau_{slow}\rangle$ normalized to that at $T = 402$ K at representative Q's. Smaller deviations from Arrhenius behavior (the straight dash line) are evident at larger Q's in both cases. Error bars represent one standard error and where not seen they are smaller than the symbol size.

limited data even with an extended time window[18,19,30,33,40,41]. Furthermore, the previous NSE results of CKN were not able to resolve reliable values of $f$ for higher temperatures, e.g., at $T > 400$ K[30,40,41].

The previous study of CKN by Tengroth et al.[47] revealed that, while the decrease in the N-N coordination number at larger distances is the same as that expected from thermal expansion, at local distances it is much larger. They attributed this to an inhomogeneous thermal expansion in the material; the nitrate-cation distances are stabilized by the Coulombic interactions and will not increase as much as nitrate-nitrate ion distances upon heating. This difference is local (it extends not far beyond the nearest nitrate-nitrate ion distance) and is expected to be averaged out at larger distances. In this regard, the decreased scattering intensity above the structure factor main peak at $Q > 1.8$ Å$^{-1}$ upon increasing temperature (Fig. 1) is likely a result of the anomalous decrease of the N-N coordination number at local distances. Therefore, our observation of a weaker temperature dependence of the dynamics (both the activation energy and the exponent $\beta$) at length scales corresponding to the ionic oxygen-cation correlations, can be explained by the stabilization mechanism of the local environment due to Coulombic interactions, not contradicting the decreasing intensity of $S(Q)$ upon increasing temperature.

The intrinsic correlations between neighboring particles may give more in-depth understanding of the microscopic relaxation dynamics of viscous liquids. A good comparison to CKN would be ZnCl$_2$, a prototypical strong ionic glass former[59], consisting of Zn(Cl$_{1/2}$)$_4$ tetrahedral motifs[65]. Similar to CKN, previous NSE

measurements on ZnCl$_2$ liquid[66] revealed different temperature dependence of the α-relaxation time at different Q. But in contrast, an anomalous increase of $\beta$ towards a simple exponential ($\beta = 1$) upon decreasing temperature was observed in ZnCl$_2$ liquid, probably related to the the formation of a network structure on cooling which favors homogeneous dynamics[66]. It has been well documented that the structure factor of CKN as observed by neutron scattering is dominated by the planar triangular nitrate ions which seem to orientationally cluster in chainlike aggregates, and the cations take up space in between the aggregates of the nitrate triangles[47]. We argue that the dynamical behavior at the length scales of >2.6 Å (i.e., $Q < 2.4$ Å$^{-1}$) reflects to a large extent the relaxations of the inter-particle correlation of the nitrate ions which gives rise to the primary peak of $S(Q)$; while at the length scales of <2.6 Å (i.e., $Q > 2.4$ Å$^{-1}$) the dynamics are dominated by the nitrate-cation (mostly oxygen-cation) correlations where the local environment is relatively more stabilized against temperature change due to the Coulombic interactions, leading to the disappearance of temperature dependence of $\beta$ and the fact that the relaxation time becomes more Arrhenius-like. The dynamic features at $Q > 2.4$ Å$^{-1}$ are not a result of the contribution from the incoherent scattering, as indicated by our polarized neutron diffraction data (detailed discussion can be found in the Supplementary Information following Supplementary Fig. 2). Interestingly, a strong temperature dependence of $\beta$ and non-Arrhenius behavior have often been observed at Q values below the main structure factor peak in multicomponent glass-forming systems, like alloy liquids, even when incoherent neutron scattering dominates[67]. Whether these effects are universal in other glass forming systems with distinct nature of interatomic/molecular interactions would be an intriguing question.

The spin-echo measurements provide detailed information on the slow α-process, however, the fast process associated with the relaxation of particles before escaping from the cages formed by the nearest neighbors is not fully covered in the spin-echo time window, as shown in Fig. 1. To access short time scales less than 1 ps, we performed quasi-elastic neutron scattering measurements for CKN on the AMATERAS spectrometer at J-PARC. The measured time-of-flight spectra were Fourier deconvoluted into the time domain ISF $I(Q,t)/I(Q,0)$. In Fig. 6 we plot representative data from the two measurements, the AMATERAS data at $T = 420$ and 480 K and the WASP data at $T = 424$ K and 483 K, at $Q = (1.26, 1.86$ and 2.60) Å$^{-1}$, respectively. Note that the 3–4 K temperature discrepancy between these two sets of data is trivial since the fast process is much less affected by temperature compared to the slow α-process. The match of the AMATERAS and the WASP data is unambiguously evident in the overlapping time range between 2.6 ps and 6 ps. We emphasize that we did not make any artificial adjustments to the data to force the match, in contrast to that done in previous studies[32,68,69]. Note that for time-of-flight measurements the neutron scattering signal is given by the sum of coherent ($I_{coh}$) and incoherent scattering ($I_{inc}$), by $I_{coh} + I_{inc}$, while for spin-echo measurements the dynamic signal is encoded in the polarization of the beam and given by $I_{coh} - 1/3I_{inc}$, where the incoherent scattering is highly suppressed. Therefore, the agreement between the AMATERAS and the WASP data also indicates that the incoherent scattering in CKN is negligible even at Q values away from the peak of $S(Q)$. As seen in Fig. 6, these combined data can be well fitted to Eq. (1) with the same parameters. The values of $\tau_{fast}$ from the fits are in the sub-picosecond time scale, which will be reported in detail elsewhere, thus not discussed in the present work.

In summary, important insights into the relaxation dynamics in the glass-forming liquid CKN have been obtained by fully mapping the Q- and T-dependence of the collective ISFs, only

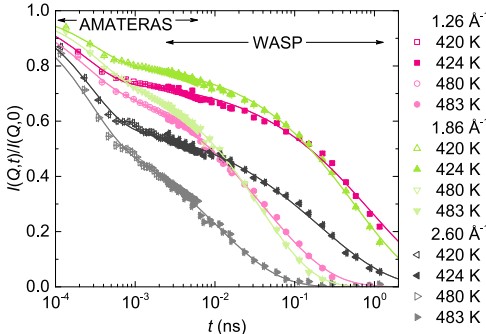

**Fig. 6 Joint intermediate scattering functions of CKN covering a broad time window at three representative $Q$'s: (1.26, 1.86, and 2.60) Å$^{-1}$.** The data presented by open symbols from $t = 0.1$ ps to 6 ps are Fourier deconvoluted time-of-flight spectra measured on AMATERAS, those presented by solid symbols from $t = 2.6$ ps to 1.2 ns are the WASP data. The AMATERAS data are measured at $T = 420$ K and 480 K, and the WASP data are at $T = 424$ K and 483 K. These two sets of data match in the overlapping time range without any artificial adjustment. The solid lines represent fit to Eq. (1) of the joint intermediate scattering functions. Error bars represent one standard error and where not seen they are smaller than the symbol size. Source data are provided as a Source Data file.

possible using the recently developed WASP neutron spectrometer at the ILL, France. Our study in CKN liquid in a wide range of temperatures and covering the full range of length scales across the structure factor peak and valley, reveals, unambiguously, a two-step relaxation. The fast exponential decay process is followed by a slow α-process of a stretched nature. The relaxation time, amplitude, and stretching exponent of the α-relaxation show remarkable $Q$-dependence and vary in phase with $S(Q)$. For all studied $Q$'s the amplitude of the α-relaxation decreases with increasing temperature and plateaus at $T > 470$ K. For $Q < 2.4$ Å$^{-1}$, the stretching exponent $β$ increases with increasing temperature and levels off at $T > 470$ K, and the temperature dependence of $τ_{slow}$ and $β$ as quantified by $D$ and $T_0$, and $dβ/dT$, respectively, follow $S(Q)$- or $-S(Q)$-like $Q$-dependence; however, for $Q > 2.4$ Å$^{-1}$, $β$ increases rather slower and eventually becomes temperature independent at higher $Q$'s, accompanied by less deviations from Arrhenius behavior of $τ_{slow}$. These contrasting behaviors indicate distinct relaxation mechanisms at the different microscopic length scales, and reveal the characteristic spatial length of ≈2.6 Å: Above 2.6 Å, the relaxation is dominated by the inter-particle correlation of the nitrate ions; while below 2.6 Å, the relaxation exhibits localized feature manifested as temperature independent heterogeneity and more Arrhenius-like behavior, due to the stabilization mechanism of the Coulombic interaction between nitrates and cations.

## Methods

**Sample preparation.** As CKN is a hydroscopic material, much care should be taken in the preparation process. The CKN sample was obtained by mixing appropriate weight of high-purity calcium nitrate tetrahydrate (Ca(NO₃)₂•4H₂O, 99.95% purity) and potassium nitrate (KNO₃, 99.995% purity) purchased from Sigma-Aldrich; the mixture was heated slowly up to above the melting point and dried in a vacuum oven at $T = 523$ K for over 48 h. The dehydrated mixture was then transferred immediately to a glove box and heated again up to $T = 523$ K for 12 h under vacuum to remove any possible water absorption from the atmosphere that may have occurred during the transfer. Then an appropriate amount (≈10 g) of molten mixture was sealed into a standard vanadium annular can (12.7 mm inner diameter and 1.2 mm thickness) in the glove box filled with high-purity helium.

**Neutron spin-echo measurements.** Wide-angle NSE measurements were performed using the high-intensity WASP spectrometer[37] at the ILL, France, which enables high-precise measurements for multiple wavevector transfer ($Q$) at the same time. To enable access to the $Q$-range from 1.08 Å$^{-1}$ to 2.82 Å$^{-1}$ which

covers the structure factor peak (at $Q = 1.86$ Å$^{-1}$) and valley of CKN, the incoming neutron wavelength λ was set to 4 Å. The data were collected at temperatures between 383 K and 519 K. The temperature was controlled with an orange cryostat ("wet" liquid He cryostat). Before measurement at each temperature, the sample was first heated to $T = 530$ K and equilibrated for 30 min, then directly cooled in an identical fashion each time to the target temperature and equilibrated for another 30 min, allowing sample equilibration and well-defined thermal history for each run. We run diffraction measurements after each data set to ensure that no crystallization occurred in the sample. All these measurements on the WASP spectrometer were finished within a total beam time of 4 days. The data were reduced by using an ILL developed software based on the commercial software package Igor Pro 8[37].

Additional NSE measurements were performed using the NGA-NSE spectrometer at the National Institute of Standards and Technology (NIST) Center for Neutron Research (NCNR)[48], USA. An incoming neutron wavelength of 5 Å was used and the measurements were performed for two $Q$'s, 0.8 Å$^{-1}$ and 1.75 Å$^{-1}$, near the pre-peak and the main peak of $S(Q)$. The software DAVE[70] was used for the NGA-NSE data reduction and the Fourier deconvolution of the AMATERAS data.

The transmissions of the sample (≈77.1%) and the empty can (≈90.7%) were measured at $T = 100$ K on the NGA-NSE spectrometer at NCNR using 5 Å neutron beam, which indicates a self-shielding factor of ≈85%, hence we estimate that double scattering would account for at most ≈15% of the measured scattering at some $Q$'s and therefore multiple scattering corrections have been deemed unnecessary.

**Time-of-flight inelastic neutron scattering measurements.** Inelastic neutron scattering measurements were carried out on the cold-neutron disk-chopper spectrometer AMATERAS[44,45] at J-PARC in Japan, using a neutron wavelength of 3.26 Å. The time of flight data of AMATERAS are processed by using the software suite Utsusemi[71]. A measurement of the empty vanadium can at $T = 300$ K was used as the instrument energy resolution for the data deconvolution. Normalized Fourier deconvoluted time-of-flight spectra were compared with the intermediate scattering function collected on the WASP spectrometer. The static structure factor $S(Q)$ was obtained by integration of the dynamic structure factor at the energy range of $[-5, 0]$ meV.

## Data availability

Source data are provided with this paper. The raw data generated in this study is accessible under https://doi.org/10.5291/ILL-DATA.DIR-202. Source data are provided with this paper.

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

## Acknowledgements

We thank Tanya J. Dax, Yamali Hernandez, Donna A. Kalteyer and Michihiro Nagao for their help on the experiments. This work is supported by the U.S. Department of Energy, Office of Science, Office of Basic Energy Sciences, Materials Sciences and Engineering Division, under Award Number DE-SC0014084. The NSE measurements on the WASP spectrometer at ILL were performed under proposal number DIR-202. Access to the NGA-NSE spectrometer at NCNR was provided by the Center for High Resolution Neutron Scattering, a partnership between the National Institute of Standards and Technology and the National Science Foundation under agreement No. DMR-1508249. The inelastic neutron scattering measurements on AMATERAS at J-PARC were performed based on the approved proposal No. 2019B0216.

## Author contributions

YZ and A.F. initiated the research. P.L. prepared the sample. P.F. performed the wide-angle NSE measurements on the WASP spectrometer at the ILL. P.L., A.F., M.K., and

K.N. performed the time-of-flight inelastic neutron scattering measurements on the AMATERAS spectrometer at J-PARC. P.L., Y. Zhai, and A.F. performed the NSE measurements on the NGA-NSE spectrometer at NCNR. P.L. analyzed the data and wrote the manuscript. V.G.S., P.F., M.H., A.F., and YZ contributed to the writing and editing of the paper. All authors engaged in useful discussions throughout the experiments and preparation of the manuscript.

## Competing interests

The authors declare no competing interests.
