## [Peer Review File · Nature Communications]

Q-dependent Collective Relaxation Dynamics of Glass-Forming Liquid Ca_{0.4}K_{0.6}(NO₃)_{1.4} Investigated by Wide- Angle Neutron Spin-EchoEditorial Note: Parts of this peer review file have been redacted as indicated to maintain the confidentiality of unpublished data.

Review report of the manuscript NCOMMS-21-37605 "Q-dependent Collective Relaxation Dynamics of

Glass-Forming Liquid $\text{Ca}_{0.4}\text{K}_{0.6}(\text{NO}_3)_{1.4}$ Investigated by Wide-Angle Neutron Spin-Echo" by Y Z and collaborators

The work is focused on the neutron scattering investigation of the nanosecond relaxation of ionic glass former $\text{Ca}_{0.4}\text{K}_{0.6}(\text{NO}_3)_{1.4}$ (CKN). CKN is well known glass former materials which consists from ion of different sizes. It is very often used a model system as a closest approximation of Lennard Jones system. The specialty of the glass transition lies in large changes in microscopic dynamics with no remarkable structural changes. Hence the glass transition is explained as a structural arrest resulting from non-linear dynamics behavior and coupling of various motion (mode coupling). Previously, strong non-Arrhenius behavior for CKN viscosity has been found by A. Angell. Two steps of relaxation process at nano and picosecond time scale has been found by F. Mezei and coworkers. In the present work the authors have concentrated on the structural relaxation mostly above 2 \AA^{-1} . They have found striking differences between the structural relaxations above and smaller $Q=2 \text{ \AA}^{-1}$ using very moderate modeling and fitting. The major findings of this work are:

1. Observation of two steps of the structural relaxation in liquid equilibrium state at high temperatures and not only for highly viscous state;
2. Different behavior of structural relaxation time and the stretched exponent β for $Q < 2.4 \text{ \AA}^{-1}$ and $Q > 2.4 \text{ \AA}^{-1}$;
3. Remarkable change in the temperature dependence of the α -relaxation amplitude that takes place at 463 K. Such kink was not observed before and is about 100 K higher as the value of critical temperature proposed before for CKN by Mode Coupling Theory.

In their work the authors fully exploited the powerful capability of neutron scattering to provide Q resolved investigation of microscopic dynamics. Also remarkable is the overlap of the spectra taken by different spectrometers for different time scales. This delivers convincing evidence that spurious scattering, like multiple scattering, has non-relevant contribution to the scattering. I think the work is novel, interesting and will benefit the general understanding of the glass transition and glassy dynamics-

However, I have following remarks:

1. One of the major aspects of the work is the Q-dependence of the parameters and its relation to the structure factor $S(Q)$. $S(Q)$ is however only shown as a small insert in one of the figures. For the benefit of the readers, I think, $S(Q)$ shall be presented on the separate figure. It would be also advisable to point out, which interatomic/ interionic correlations cause the peaks in $S(Q)$ and introduce in text the different length scales which exist in the glass forming systems.
2. Authors use the notion of liquid CKN below 400 K as supercooled, which is wrong. It is common to call supercooled liquids if they remain in the liquid state below the freezing /crystallization temperature. Since for glasses we do not observe crystallization the glass transition temperature, e.g. where the viscosity reaches some certain value, is usually used. It is commonly accepted to call liquids in the temperature range between the temperature of the equilibrium liquid and the glass transition temperature as highly viscous liquids or melts.
3. All parameters in the fit of the Eq. 2 are Q-dependent, which could explain why the fitting was working so well. Was some effort done to reduce Q dependence, for instance, set T constant? For

this purpose the commonly accepted glass transition temperature of 333 K or 340 K from which was determined from fitting of the shear relaxation time be used. What happens in this case?

4. The title include "Collective relaxation dynamics". Does the author has evidences that this relaxation is collective/cooperative? The fact that CKN has mostly coherent scattering cross section does not necessarily mean that the observed process is the cooperative one.
5. The work uses many parameters / processes introduced first by Mode Coupling Theory, therefore I'd suggest to briefly introduce this work. Some classical work are: E. Leutheusser, *Phys. Rev. A* 29, 2765 (1984); U. Bengtzelius, W. Gotze, and A. Sjolander, *J. Phys. C* 17, 5915 (1984); S. P. Das, G. F. Mazenko, and S. Ramaswamy, and F. F. Toner, *Phys. Rev. Lett.* 54, 118 (1985)
6. The figures 3 c) and 3d) have no agenda for color coding. I also think that there are too many different color points, which makes sometime the reading of the figures quite difficult. The authors might reduce the number of the curves shown for better understanding.

To conclude: based on the novel and quality of the results I strongly recommend the presented manuscript for publications with some modifications along the lines suggested above.

Reviewer #2 (Remarks to the Author):

Although our understanding of the dynamics of glass-forming liquids has advanced well in the last decades, there are still many open questions and therefore this topic is an attractive domain of research. One bottleneck that experiments are facing is the access to the dynamics on the scale of the atoms since so far neutron scattering techniques offered only a limited access time and length-scale, i.e. wave-vector. The latest generation of detectors allows now to make (finally) a significant step forward in that one can now measure the time correlation functions over several decades in time and this with an excellent accuracy. This progress in instrumentation is exploited by the authors of the present manuscript in that they analyze the wave- and temperature dependence of the relaxation dynamics of CKN, a simple ionic glass-former. The presented data is of very high quality and reveal interesting effects. Unfortunately I find that the discussion of the data not very satisfactory (see below for details) and hence advise the authors to improve this part before the manuscript can be accepted for publication.

- 1) The list of authors include "Y Z": I suppose that this should read "Yang Zhang", no?
- 2) Abstract and several other places in the text: The authors make reference to "the valley of the structure factor". It is more standard to refer to "the first minimum of the structure factor".
- 3) p.2: "In recent years...": The cited papers are 20 years old, which shows that this is not really "recent".
- 4) p.2: "Previous studies...": It might be fair to cite at this place also some simulation studies and experiments on colloidal systems that have been done to investigate dynamical heterogeneities.
- 5) p.2: "...have revealed that in the intermediate supercooled liquid region, a transition from the slow alpha-relaxation to the Johari-Goldstein (JG) beta-relaxation as the temperature is decreased." I'm not sure to understand this statement. There is no transition between these two relaxation processes. The only thing that happens is that at intermediate temperatures the alpha-process masks the JG process while at low T the JG process becomes visible since the alpha process has moved to lower frequencies. So the authors should make this clear.
- 6) p.3: The experiment accesses the neutron scattering intensity $I(Q,t)$. The latter is the weighted sum of the partials. So it would be useful if the authors would tell us the contribution of the various pairs (say in the SI) so that one understands what type of dynamics is really measured.
- 7) Fig 1b indicates that at none of the temperatures considered the system is in a "normal" liquid state, i.e. the correlation function decays in an exponential manner. In other words, for all T's the liquid is glassy. Note that the fact that the highest T is above the liquidus temperature is irrelevant since, e.g., silica is also a glassy liquid at 30% *above* its melting temperature.

Questions:

i) Is it possible to estimate the onset temperature, i.e. the T at which the liquid crosses over from normal to glassy? From the data I expect it to be around 500K. ii) Is there any estimate in the literature of T_{MCT} , the critical temperature of mode-coupling theory?

8) In Fig. 2a the authors show also $S(Q)/Q^2$. Why should this be a reasonable estimate for the alpha-relaxation time? The de Gennes narrowing is an effect that is obtained from the *short* time expansion of the time-correlation function. In other words, it applies to τ_{fast} and not τ_{slow} . Of course it might be that the Q -dependence of τ_{slow} inherits some of the Q -dependence of τ_{fast} , but there is no theory which tells that this is indeed the case. So the authors might want to formulate this in a more careful manner. (And yes, this misleading reasoning is found often in the literature, but one should not propagate it.)

9) p.5: "...leading to increased dynamic heterogeneity as manifested by the decreased value of beta." One has to be careful with such statements: An increased dynamical heterogeneity will result in a decrease of beta. However, a decrease of beta does not necessarily signal an increased dynamical heterogeneity since the particle based relaxation process might have become more stretched.

10) p.6: "It has been revealed in polybutadiene and ortho-terphenyl that the transition from the non-Arrhenius alpha-process to the JG beta-process which follows Arrhenius behavior, occurs in the supercooled liquid region in the Q -range above the primary peak of $S(Q)$ ". As already mentioned under #5), I really do not see why one should invoke here the JG beta process. There is no evidence in the data that one really sees this process. Note that in the frequency domain the beta-process has normally a much smaller amplitude than the alpha process and this translates in a small value of the corresponding Debye-Waller factor f . So where in the present data is the evidence that there is indeed a JG beta process?

Quite a bit of text in the manuscript discusses this process but I have the impression that there is a lot of speculation and no real evidence. So I invite the authors to present this evidence or (perhaps better) remove most of these speculations.

11) p. 8: "...and further demonstrate that a constant f indeed exists but only in the normal liquid state, which was not established from the previous results of CKN". As mentioned above, Fig. 1

indicates that for all temperatures studied here the sample is in the glassy state, i.e. it is not a normal liquid. Note that in the normal liquid state the non-ergodicity parameter f is zero since the shoulder/plateau in $I(Q,t)$ has disappeared. So the authors might want to adapt the text.

12) p.8: "...where the incoherent scattering is highly suppressed." Why is this so?

13) Fig. 3a (right panel): Are these error bars realistic? It is strange that the data points line up very well on a horizontal line while the error bars are way larger than the typical deviations from this line.

14) Fig. 3a: Why is the data for $Q=0.80$ that much higher than the one for the other wave-vectors? Also the last data point for $Q=1.26$ seems to be way off the trend suggested by the points at higher T . Why is this?

In summary: The authors have determined with excellent precision the Q and T dependence of the coherent intermediate scattering function of a simple glass-former. They give good evidence that the relaxation dynamics of this system shows a crossover at around $T_X=470\text{K}$. The authors try to connect this temperature to the existence of the JG peak, but for me the connection is extremely handwavy and thus not convincing (also because the transition looks quite sharp and this is not what would be expected from the very broad JP peak). One possible alternative might (!) be that this cross-over is related to the mode-coupling temperature, i.e. above T_X one has the normal MCT dynamics in which the height of the non-ergodicity parameter is constant and below T_X one has the increase of f as expected from theory. Thus T_X is just the temperature at which the dynamics changes from flow-like (as proposed by the ideal version of MCT) to hopping like (generalized MCT). Thus the presented results are indeed coherent with a simple cross-over scenario predicted by theory. I certainly do not want to force the authors to interpret their beautiful data by means of this theory, but they should at least think about in this direction and/or come up with a better discussion of the meaning of the data. The present version of the manuscript is lacking a convincing discussion and hence I invite the authors to improve this point.

Reviewer #3 (Remarks to the Author):

This manuscript reports on a neutron scattering investigation of the relaxation processes in the fragile glass-former CKN in the normal and supercooled liquid state. The study covers about 4 decades in time between a fraction of ps and few ns, and a wide-angle Q -range between 0.8 and 2.8 \AA^{-1} .

The same sample was already studied with this approach as reported in a series of papers appeared in the 80s and 90s: this is carefully discussed in the manuscript. While the quality of the new set of data is undoubtedly superior than in those previous studies, by choosing an extensively studied sample the authors have decided to face here the challenge of extracting qualitatively new results.

The main point the authors make here is that there would exist a

characteristic length of 2.6 Å below which the structural relaxation would become more Arrhenius-like and would display a temperature independent heterogeneity. The basis of these claims is in Fig.3.

I observe that the fit results for beta (Fig.3a) and for tau (Fig.3b) look extremely well aligned as compared to the reported error bar (one standard error, as reported by the authors). It would be important to provide more detail about the fitting procedure that has been used.

Have all parameters been left free? How have been the error bars estimated? In fact, when the authors fit these (fit) results with a linear temperature dependence (for the beta) or with the VFT law (for the tau), they obtain very precise values of the parameters (D , T_0 and $d(\text{beta})/dT$).

They also obtain values higher than T_g for several T_0 values

corresponding to fits of tau(Q) data at low-Q's. How do the authors interpret that? Do they believe that the VFT model is still adequate to describe these data sets? and then what does the observation of a change of the fitted parameters (D , T_0) of the VFT law really mean?

More on the technical side: did they investigate possible correlations between D and T_0 ? How would the chi-squared of the fits of the VFT-law to the tau(Q) data change if both parameters would be fixed to the values found at intermediate Q's?

The same questions hold for the data for beta.

Other points:

i) How do the authors relate the disappearance of the temperature dependence of the stretching parameter to the fact that the relaxation time becomes more Arrhenius-like? I understand that this is their observation, but do the authors have an explanation for that?

ii) Fig.S2c shows that the shape of the tau(T) curve changes with Q. I would like the authors to discuss that more in detail.

In fact, the same trend is not clear on the original data plotted in Fig.S2b together with the error bars. Moreover, a VFT law has of course a temperature-dependent activation energy. This implies that different portions of it, corresponding to different times, necessarily display a different curvature. Can the authors

exclude this scenario?

iii) It would be useful to show the results of Fig.2a in linear scale, maybe normalizing the $\tau(Q)$ data collected at the different temperatures. Would the comparison to $S(Q)/Q^2$ remain convincing even in this representation?

iv) pag.2, second paragraph, 7th line - something is missing in that sentence.

On the basis of these considerations, I believe that while the main claims of the authors are potentially interesting, the effects they are behind are very small, and the presentation should be strengthened to clearly demonstrate that they are robust in fact against fitting uncertainties.

Manuscript ID: NCOMMS-21-37605

Title: **"Q-dependent Collective Relaxation Dynamics of Glass-Forming Liquid Ca_{0.4}K_{0.6}(NO₃)_{1.4} Investigated by Wide-Angle Neutron Spin-Echo"** by P. Luo, et al.

Response letter to the comments of the Referees

*We thank the Referees for their careful reading of our manuscript and for their valuable comments and suggestions. Particularly, we thank the Referees for raising the concerns regarding the error bars. In fact, we realized that for the old version of the manuscript the calculation of error bar was inappropriate, therefore, we reanalyzed all the data, which does not change the original conclusions but gives more reasonable error bars. Following the suggestions of the Referees, in the revised manuscript we added a figure reporting the static structure factors at different temperatures as well as additional discussions based on the microscopic structural correlations. To improve readability and to better highlight the important information at the same time, we moved some data in Fig. 3 of the old manuscript to the Supplementary Information. We reproduce below the original comments of the Referees marked in **bold**, followed by our point-by-point reply. We have also modified the text of the manuscript accordingly and hope the Referees find the present version suitable for publication in Nature Communications.*

Report of Reviewer #1

(From the referee) Reviewer #1 (Remarks to the Author): Based on the novel and quality of the results I strongly recommend the presented manuscript for publications with some modifications along the lines suggested in the attachment.

Review report of the manuscript NCOMMS-21-37605 "Q-dependent Collective Relaxation Dynamics of Glass-Forming Liquid $\text{Ca}_{0.4}\text{K}_{0.6}(\text{NO}_3)_{1.4}$ Investigated by Wide-Angle Neutron Spin-Echo" by Y Z and collaborators

The work is focused on the neutron scattering investigation of the nanosecond relaxation of ionic glass former $\text{Ca}_{0.4}\text{K}_{0.6}(\text{NO}_3)_{1.4}$ (CKN). CKN is well known glass former materials which consists from ion of different sizes. It is very often used a model system as a closest approximation of Lennard Jones system. The specialty of the glass transition lies in large changes in microscopic dynamics with no remarkable structural changes. Hence the glass transition is explained as a structural arrest resulting from non-linear dynamics behavior and coupling of various motion (mode coupling). Previously, strong non-Arrhenius behavior for CKN viscosity has been found by A. Angell. Two steps of relaxation process at nano and picosecond time scale has been found by F. Me

-1

⁻¹ using very moderate modeling and fitting. The major findings of this work are:

- 1. Observation of two steps of the structural relaxation in liquid equilibrium state at high temperatures and not only for highly viscous state;**
- 2. Different behavior of structural relaxation time and the stretched exponent β
^{-1 -1};**
- 3. Remarkable change in the temperature dependence of the α - relaxation amplitude that takes place at 463 K. Such kink was not observed before and is about 100 K higher as the value of critical temperature proposed before for CKN by Mode Coupling Theory.**

In their work the authors fully exploited the powerful capability of neutron scattering to provide Q resolved investigation of microscopic dynamics. Also remarkable is the overlap of the spectra taken by different spectrometers for different time scales. This delivers convincing evidence that spurious scattering, like multiple scattering, has non-relevant contribution to the scattering. I think the work is novel, interesting and will benefit the general understating o the glass transition and glassy dynamics-

Reply: We appreciate the positive comments and constructive suggestions provided by the Referee which are very helpful for improving our manuscript. In the following, we provide a

point-by-point reply to the Referee's remarks, and we believe that all the points raised have been satisfactorily addressed.

However, I have following remarks:

(From the referee) 1. One of the major aspects of the work is the Q-dependence of the parameters and its relation to the structure factor $S(Q)$. $S(Q)$ is however only shown as a small insert in one of the figures. For the benefit of the readers, I think, $S(Q)$ shall be presented on the separate figure. It would be also advisable to point out, which interatomic/interionic correlations cause the peaks in $S(Q)$ and introduce in text the different length scales which exist in the glass forming systems.

Reply: We have presented $S(Q)$ at different temperatures in a separate figure (new Figure 1) in the revised manuscript and added discussions in the text related to the different correlation length scales in the system. Also, additional discussions are made for the length scale dependent relaxation dynamics based on the microscopic structural correlations. The reader is also directed to the relevant Ref. 47 (Tengroth et al.) which reported the contribution of the different partial structure factors to the neutron scattering diffraction pattern

(From the referee) 2. Authors use the notion of liquid CKN below 400 K as supercooled, which is wrong. It is common to call supercooled liquids if they remain in the liquid state below the freezing /crystallization temperature. Since for glasses we do not observe crystallization the glass transition temperature, e.g. where the viscosity reaches some certain value, is usually used. It is commonly accepted to call liquids in the temperature range between the temperature of the equilibrium liquid and the glass transition temperature as highly viscous liquids or melts.

Reply: In the revised manuscript, according to the Referee's suggestion, we have replaced "supercooled liquid" by "viscous liquid" or "highly viscous liquid" for CKN. However, we would like to respectfully point out that CKN can indeed freeze at a temperature of $T_m = 483$ K [J. Phys. Chem. 95, 7050 (1991)], and, therefore following the convention, CKN at temperatures between T_m and $T_g = 333$ K could be referred to as supercooled.

(From the referee) 3. All parameters in the fit of the Eq. 2 are Q-dependent, which could explain why the fitting was working so well. Was some effort done to reduce Q dependence, for instance, set T constant? For this purpose the commonly accepted glass transition temperature of 333 K or 340 K form which was determine from fitting of the shear relaxation time be used. What happens in this case?

Reply: We did attempt to fit the data by setting T_0 constant. The following two upper plots show the obtained results an example at $Q = 2.60 \text{ \AA}^{-1}$ where the blue lines are fit with both T_0 and D free, and the red lines are the best fit with T_0 or D fixed at the values obtained from the macroscopic shear relaxation time ($T_0 = 340 \text{ K}$, $D = 1.53$). It is evident that fixing either T_0 or D prevents to fit the data satisfactorily. Fixing T_0 at 333 K also results in an unsatisfactory fit of the data. These two plots are also reported in the Supporting Information.

This result can also be appreciated directly without any VFT fit. The following lower panels show the normalized relaxation time in logarithmic scale. In agreement with the VFT fit results, smaller deviations from the Arrhenius behavior (black dash line) are evident at larger Q 's, in both the relaxation time τ_{slow} (left panel) and the average relaxation time $\langle \tau_{\text{slow}} \rangle$ (right panel), reinforcing the conclusion that the temperature dependence of the relaxation time is different for different Q -range. These two plots are also shown in Fig. 5 of the new manuscript.

[redacted]

(From the referee) 4. “C” D has evidences that this relaxation is collective/cooperative? The fact that CKN has mostly coherent scattering cross section does not necessarily mean that the observed process is the cooperative one.

Reply: The polarized $S(Q,0)$ which is used for the normalization of the echo signal is positive for all Q 's studied here. Since coherent scattering gives positive polarization and incoherent scattering gives rise to negative polarization, having positive $S(Q,t)$ functions decaying down (and not up) shows that the dynamics detected comes from coherent scattering. Coherent neutron scattering ISFs comprise contributions from both a single particle and distinct contribution, resulting in what is commonly referred to as the collective dynamics. This is an important distinction as many dynamic neutron scattering studies focus on the incoherent scattering and therefore are sensitive to the single particle dynamics only. On the other hand, we agree with the referee that the coherent ISF does not necessarily reflect a cooperative motion. Therefore, when introducing the collective ISF, a note has been added stating: “The coherent ISF is comprised of both the self and distinct component, and therefore reflect motions that are not necessarily cooperative.”

(From the referee) 5. The work uses many parameters / processes introduced first by Mode C , ’ to briefly introduce this work. Some classical work

are: E. Leutheusser, Phys. Rev. A 29, 2765 (1984); U. Bengtzelius, W. Gotze, and A. Sjolander, J. Phys. C 17, 5915 (1984); S. P. Das, G. F. Mazenko, and S. Ramaswamy, and F.

F; Toner, Phys. Rev. Lett. 54, 118 (1985)

Reply: The Mode-Coupling Theory has been appropriately introduced in the revised manuscript referencing these classic works.

(From the referee) 6. The figures 3 c) and 3d) have no agenda for color coding. I also think that there are too many different color points, which makes sometime the reading of the figures quite difficult. The authors might reduce the number of the curves shown for better understanding.

Reply: We have moved some curves to the Supplementary Information and show only representative curves in the manuscript to highlight the important information and make it easier for the reader to discern the different curves.

To conclude: based on the novel and quality of the results I strongly recommend the presented manuscript for publications with some modifications along the lines suggested above.

Reply: We thank the Referee again for his/her strong recommendation for publication of our manuscript. We believe that the Referee's concerns have been addressed either in the above point-by-point reply or in the revised manuscript.

Report of Reviewer #2

(From the referee) Reviewer #2 (Remarks to the Author): Although our understanding of the dynamics of glass-forming liquids has advanced well in the last decades, there are still many open questions and therefore this topic is an attractive domain of research. One bottleneck that experiments are facing is the access to the dynamics on the scale of the atoms since so far neutron scattering techniques offered only a limited access time and length-scale, i.e. wave-vector. The latest generation of detectors allows now to make (finally)

a significant step forward in that one can now measure the time correlation functions over several decades in time and this with an excellent accuracy. This progress in instrumentation is exploited by the authors of the present manuscript in that they analyze the wave-and temperature dependence of the relaxation dynamics of CKN, a simple ionic glass-former. The presented data is of very high quality and reveal interesting effects. Unfortunately I find that the discussion of the data not very satisfactory (see below for details) and hence advise the authors to improve this part before the manuscript can be accepted for publication.

Reply: The Referee's positive comments on our work are highly appreciated, and the suggestions about our discussion of the data have been very helpful to improve our manuscript. In the following, we provide a point-by-point reply to the Referee's remarks and detail how the manuscript has been modified accordingly. We believe that the points raised by the referee have all been addressed and hope that the Referee think that the revised manuscript can be accepted for publication.

(From the referee) 1) The list of authors include "Y Z": I suppose that this should read "Yang Zhang", no?

Reply: "Yang Zhang" is in the middle of changing his name to "Y Z" officially.

(From the referee) 2) Abstract and several other places in the text: The authors make reference to "the valley of the structure factor". It is more standard to refer to "the first minimum of the structure factor".

Reply: As shown in the new Fig. 1 of the revised manuscript, the structure factor of CKN shows a pre-peak at $Q = 0.8 \text{ \AA}^{-1}$, which makes a minimum before the structure factor main peak at $Q \approx 1.86 \text{ \AA}^{-1}$; therefore, we changed the text to "the first minimum above the structure factor main peak" in the revised manuscript.

(From the referee) 3) p.2: "In recent years...": The cited papers are 20 years old, which shows that this is not really "recent".

Reply: The words "In recent years" have been changed to "Previously" accordingly.

(From the referee) 4) p.2: "Previous studies...": It might be fair to cite at this place also some simulation studies and experiments on colloidal systems that have been done to investigate dynamical heterogeneities.

Reply: The following representative papers on simulation studies and experimental studies of colloidal systems that investigated dynamic heterogeneities have been referenced in the revised manuscript.

1. Widmer-Cooper, A. & Harrowell, P. Predicting the Long-Time Dynamic Heterogeneity in a Supercooled Liquid on the Basis of Short-Time Heterogeneities. *Phys. Rev. Lett.* **96**, 185701 (2006).
2. Kob, W., Donati, C., Plimpton, S. J., Poole, P. H. & Glotzer, S. C. Dynamical Heterogeneities in a Supercooled Lennard-Jones Liquid. *Phys. Rev. Lett.* **79**, 2827–2830 (1997).
3. Kob, W., Roldán-Vargas, S. & Berthier, L. Non-monotonic temperature evolution of dynamic correlations in glass-forming liquids. *Nat. Phys.* **8**, 164–167 (2012).
4. Berthier, L. *et al.* Direct Experimental Evidence of a Growing Length Scale Accompanying the Glass Transition. *Science*. **310**, 1797–1800 (2005).
5. Weeks, E. R., Crocker, J. C., Levitt, A. C., Schofield, A. & Weitz, D. A. Three-Dimensional Direct Imaging of Structural Relaxation Near the Colloidal Glass Transition. *Science*. **287**, 627–631 (2000).

(From the referee) 5) p.2: "...have revealed that in the intermediate supercooled liquid region, a transition from the slow alpha-relaxation to the Johari-Goldstein (JG) beta-relaxation as the temperature is decreased." I'm not sure to understand this statement. There is no transition between these two relaxation processes. The only thing that happens is that at intermediate temperatures the alpha-process masks the JG process while at low T the JG process becomes visible since the alpha process has moved to lower frequencies. So the authors should make this clear.

Reply: Generally, in *macroscopic* measurements such as dielectric spectroscopy and dynamic mechanical analysis, one can observe a decoupling of the relaxation from a single mode into the sum of an Arrhenius type JG β process and the α process that continues following the non-Arrhenius behavior upon cooling below some temperature. This decoupling temperature is usually not directly observed but evinced by extrapolating the data from lower temperatures towards a crossover. Saito et al. [PRL 109, 115705 (2012); J. Chem. Phys. 140, 144906 (2014)] employed the nuclear resonance X-ray scattering technique to study the relaxation dynamics of glass formers and *directly* observed a crossover from a regime dominated by the slow α to one dominated by the JG β process; this crossover occurring in the time scale of ≈ 10 ns to $\approx 10^3$ ns. More intriguingly, with the ability to resolve the length scale dependence, they found that this crossover occurs only in the Q -range of the first minimum above the primary peak of $S(Q)$, not around the primary peak, and “bifurcation” of the α -process into the JG β -process and the α -process, as often observed in macroscopic measurements, does not occur at certain Q 's. It means that both the α -process and the JG β -process do not co-exist if the observation is set to a specific length scale. Therefore, they used the terminology of “transition” to formulate this dynamic phenomenon.

Due to the limited study on the α and JG β crossover phenomenon with Q dependence, the above-mentioned argument by Saito et al. has not been widely accepted; therefore, we replaced “transition” with “crossover” to avoid misunderstanding. We have clarified this point in the revised manuscript.

(From the referee) 6) p.3: The experiment accesses the neutron scattering intensity $I(Q,t)$. The latter is the weighted sum of the partials. So it would be useful if the authors would tell us the contribution of the various pairs (say in the SI) so that one understands what type of dynamics is really measured.

Reply: The following table shows the neutron scattering cross sections (σ) for different elements present in CKN, adapted from <https://www.nist.gov/ncnr/planning-your-experiment/scattering-length-periodic-table>. We see N and O have much larger neutron scattering cross section than Ca and K. In addition, the concentrations of N and O are much larger than Ca and K in CKN. Therefore, the neutron scattering signals in CKN are to a large extent from N and O.

	σ_{coherent} (barn)	$\sigma_{\text{incoherent}}$ (barn)
N	11.01	0.5
O	4.232	0.0008
Ca	2.78	0.05
K	1.69	0.27

We have included the static structure factors at different temperatures in the new Figure 1 of the manuscript, see also below. The contributions of the partial structure factor as observed by neutron scattering for CKN have been well documented by Tengroth et al. [*Phys. Rev. B* **64**, 224207 (2001)] (Ref. 47 in the new manuscript). Basically, the pre-peak at 0.8 \AA^{-1} could arise from the chainlike aggregates formed by orientationally correlated neighboring nitrate ions such that the oxygen atoms are facing each other, the primary peak is to a large extent the structure of the nitrate ions, and the third peak at 2.85 \AA^{-1} manifests mainly the ionic oxygen-cation correlations.

We have also added the above information in the manuscript and SI, and more discussions related to the structure and dynamics in the text.

(From the referee) 7) Fig 1b indicates that at none of the temperatures considered the system is in a "normal" liquid state, i.e. the correlation function decays in an exponential manner. In other words, for all T's the liquid is glassy. Note that the fact that the highest T is above the liquidus temperature is irrelevant since, e.g., silica is also a glassy liquid at 30% *above* its melting temperature.

Questions: i) Is it possible to estimate the onset temperature, i.e. the T at which the liquid crosses over from normal to glassy? From the data I expect it to be around 500K.

ii) Is there any estimate in the literature of T_{MCT} , the critical temperature of mode-coupling theory?

Reply: Thanks for pointing this out. Even at 519 K which is the highest temperature studied, the intermediate scattering function is stretched rather than simply exponential. So, we agree with the Referee that the dynamics are glassy instead of "liquid-like" in all the studied temperatures herein. Moreover, the relaxation time at all Q 's, as well as the viscosity data in literature, can be fitted with the non-Arrhenius VFT law in our studied temperature range. Thus, it is difficult to estimate the onset temperature in the present study.

As for T_{MCT} , Mezei, et al. [Phys. Scr. T19B, 363–368 (1987)] did estimate it to be 368 K, but they also noticed that the measured effective nonergodicity parameter does not level off even at 403 K. The relevant figure adapted from Mezei's paper is reported below on the left, for convenience. Our data enabling the estimation of the effective nonergodicity parameter f till as large a temperature as 519 K, agree with Mezei's data and reveal that f keeps decreasing *linearly* with increasing temperature and levels off only above 470 K (see the following figure on the right, where Mezei's data was also included as open magenta diamonds).

Fig. 3. Temperature variation of the measured effective nonergodicity parameter f_{eff} at $q_0 = 1.86 \text{ \AA}^{-1}$. For comparison, the dashed line indicates the qualitative behaviour predicted theoretically with $T_0 = 95^\circ\text{C}$. (Not a fit!) The horizontal section of the dashed line above T_0 represents f_{eff}^c , the nonergodic fraction at T_0 , which is taken to be equal to the relative amplitude A of the second relaxation step, as shown in Fig. 2. The subscript “eff” indicates that the results were obtained with the phonon like contributions approximately eliminated by our apparatus.

Overall, we agree with the Referee that the usage of “normal liquid” is inappropriate in our studied temperature range, and we have removed it from the manuscript.

(From the referee) 8) In Fig. 2a the authors show also $S(Q)/Q^2$. Why should this be a reasonable estimate for the alpha-relaxation time? The de Gennes narrowing is an effect that is obtained from the *short* time expansion of the time-correlation function. In other words, it applies to τ_{fast} and not τ_{slow} . Of course it might be that the Q -dependence of τ_{slow} inherits some of the Q -dependence of τ_{fast} , but there is no theory which tells that this is indeed the case. So the authors might want to formulate this in a more careful manner. (And yes, this misleading reasoning is found often in the literature, but one should not propagate it.)

Reply: We appreciate that the Referee pointed out our inappropriate description about the de Gennes narrowing, and we have modified our manuscript accordingly to avoid propagating this misleading reasoning: we removed the related sentence in the abstract and conclusion parts, and stress that the de Gennes narrowing is an effect related to the *short* time dynamics instead of the slow processes, thus, more theoretical efforts would be needed to rigorously explain our observations on the time scale of the α -relaxation.

(From the referee) 9) p.5: “...leading to increased dynamic heterogeneity as manifested by the decreased value of beta.” One has to be careful with such statements: An increased dynamical heterogeneity will result in a decrease of beta. However, a decrease of beta does

not necessarily signal an increased dynamical heterogeneity since the particle based relaxation process might have become more stretched.

Reply: We agree that it is not rigorously correct, relaxation with a stretched exponent (<1) does not necessarily reflect dynamic heterogeneity. We have modified the above-mentioned sentence to "...leading to a wider distribution of the relaxation processes as manifested by the decreased value of", as well as the similar descriptions at other places across the manuscript.

(From the referee) 10) p.6: "It has been revealed in polybutadiene and ortho-terphenyl that the transition from the non-Arrhenius alpha-process to the JG beta-process which follows Arrhenius behavior, occurs in the supercooled liquid region in the Q-range above the primary peak of $S(Q)$ ". As already mentioned under #5), I really do not see why one should invoke here the JG beta process. There is no evidence in the data that one really sees this process. Note that in the frequency domain the beta-process has normally a much smaller amplitude than the alpha process and this translates in a small value of the corresponding Debye-Waller factor f . So where in the present data is the evidence that there is indeed a JG beta process?

Quite a bit of text in the manuscript discusses this process but I have the impression that there is a lot of speculation and no real evidence. So I invite the authors to present this evidence or (perhaps better) remove most of these speculations.

Reply: The present data do not evidence the JG β process since we are measuring at higher temperature (or shorter time scale) than that of the occurrence of the slow α and JG β crossover as shown in the figure below [see more details in Fig. S5(b) in the new manuscript]. The descriptions in the old manuscript may have been inappropriate and mislead the readers to think that we were observing the JG β process. We have reworded the related statements in the manuscript accordingly, which we think made our understanding much clearer now.

The reasons why we invoke the JG β process in the present work are that: 1) In the introduction part, we mention the reported distinct behavior of the slow α and JG β crossover at different length scales to highlight the importance of Q -dependence study of the microscopic relaxation dynamics. 2) In the part pointed out by the Referee, we think it is necessary to clarify that we are observing a more Arrhenius-like behavior at the first minimum above the primary peak of $S(Q)$ even well above the temperature of the slow α and JG β crossover. Otherwise, one may think our

observation of the more Arrhenius-like behavior at the local length scales may contain the influence of the JG β process which also occurs only at the local length scales and shows an Arrhenius behavior. But we agree with the referee that the sentence, “This Q -dependent behavior suggests that the α - to JG β -relaxation transition might be related to the change of the dynamic behavior at the local length scales prior to the apparent α - β bifurcation on cooling.” is a speculation and has been removed from the manuscript.

(From the referee) 11) p. 8: "...and further demonstrate that a constant f indeed exists but only in the normal liquid state, which was not established from the previous results of CKN". As mentioned above, Fig. 1 indicates that for all temperatures studied here the sample is in the glassy state, i.e. it is not a normal liquid. Note that in the normal liquid state the non-ergodicity parameter f is zero since the shoulder/plateau in $I(Q,t)$ has disappeared. So the authors might want to adapt the text.

Reply: Thanks for reminding, we have modified the related text across the manuscript.

(From the referee) 12) p.8: "...where the incoherent scattering is highly suppressed." Why is this so?

Reply: In general neutron scattering experiments, such as time-of-flight measurements, unpolarized neutrons are used and the neutron scattering signal is given by the sum of coherent (I_{coh}) and incoherent scattering (I_{inc}), by $I_{\text{coh}} + I_{\text{inc}}$. While for spin-echo measurements, polarized neutrons are used. Incoherent scattering changes the polarization to $-1/3$ and thus the total

polarized signal is given by $I_{\text{coh}} - 1/3I_{\text{inc}}$, which means 2/3 of the incoherent scattering is eliminated. Therefore, the incoherent scattering is highly suppressed in neutron spin-echo measurements. More details can be obtained from this recent review paper [Nat. Rev. Phys. 2, 103–116 (2020)] or the references therein.

(From the referee) 13) Fig. 3a (right panel): Are these error bars realistic? It is strange that the data points line up very well on a horizontal line while the error bars are way larger than the typical deviations from this line.

Reply: We thank the Referee's for raising this important point. We rechecked our data and our previous fitting procedures, and we found that the error bars were indeed not properly calculated. We used the standard nonlinear fitting procedure in OriginLab to perform the fittings. Basically, we noticed that the error bars of the intermediate scattering functions are very small, but that of the fitting parameters, f , τ , and σ are large; on the other hand, the error bars of τ and σ are large, but the error bars of the parameters obtained from the fit to τ and σ are not that large. This is because in our previous fittings the parameter's standard errors do not reflect the magnitude of weight, which is $1/\sigma_i^2$, where σ_i is the error bar sizes of the data being fitted. More explanations can be found here <https://www.originlab.com/doc/Quick-Help/Chi-Se-Remain>.

We have reanalyzed **all** the data in our work using the more appropriate settings ensuring that the parameter's standard errors reflect the magnitude of the weight. The results are not significantly changed, but more reasonable error bars are obtained. In the reanalyzed data, the error bars of f , τ , and σ are much smaller than before because of the high accuracy of the measured intermediate scattering functions with very small error bars. Since the error bars of f , τ , and σ get smaller, the parameters obtained from the fit to τ and σ with new error bars are not significantly changed compared with the previous ones without considering the magnitude of the weight.

(From the referee) 14) Fig. 3a: Why is the data for Q=0.80 that much higher than the one for the other wave-vectors? Also the last data point for Q=1.26 seems to be way off the trend suggested by the points at higher T. Why is this?

Reply: For this question, we guess that the Referee is referring to Fig. 3b instead of Fig. 3a.

As shown in the new Figure 1, there exists a pre-peak on the structure factor at $Q = 0.80 \text{ \AA}^{-1}$, which was suggested to arise from the chainlike aggregates formed by orientationally correlated neighboring nitrate ions based on neutron diffraction and reverse Monte Carlo modeling [Phys. Rev. B 64, 224207 (2001)]. This intermediate-range ordering could account for the larger α -relaxation time for $Q = 0.80 \text{ \AA}^{-1}$ than the other Q 's.

As for the last data point for $Q = 1.26 \text{ \AA}^{-1}$, its deviation from the fit is because of the much larger error bar size of the last two data points compared to the data at high temperatures. From the figure below we can see that the error bar size of the relaxation time at high temperatures are very small (much smaller than the size of the symbol), while that of the last two data points is remarkably larger. Therefore, when fitting the data with the instrumental weight method (weight factor = $1/\sigma_i^2$ where σ_i is the error bar sizes of the data being fitted), much smaller weights are added to the last two data points, in addition, the relaxation time here increases faster than at higher temperatures, making the fit deviate from this last point, as shown by the red line in the figure below. The much larger error bar for the last data point is because at this temperature and at this Q the relaxation time is longer than the time window of our WASP measurements, thus, the fit to the intermediate scattering function gives larger error bar for the fitted τ_{slow} . If we fit τ_{slow} without weight, i.e., not considering the error bar of each data point, the fit can capture all the data points, as shown by the black line, and the obtained VFT parameters D and T_0 , are even closer to the values obtained from the macroscopic shear relaxation time ($D = 1.53 \pm 0.15$ and $T_0 = 340.0 \pm 3.5 \text{ K}$), thus not affecting our conclusions. For the data at the other Q 's, the fit with or without weight has little effect. Therefore, we believe that for $Q = 1.26 \text{ \AA}^{-1}$ the fit without weight is more reasonable than that of adding much biased weight to the last two data points.

(From the referee) In summary: The authors have determined with excellent precision the Q and T dependence of the coherent intermediate scattering function of a simple glass-former. They give good evidence that the relaxation dynamics of this system shows a crossover at around $T_X=470\text{K}$. The authors try to connect this temperature to the existence of the JG peak, but for me the connection is extremely handwavy and thus not convincing (also because the transition looks quite sharp and this is not what would be expected from the very broad JP peak). One possible alternative might (!) be that this cross-over is related to the mode-coupling temperature, i.e. above T_X one has the normal MCT dynamics in which the height of the non-ergodicity parameter is constant and below T_X one has the increase of f as expected from theory. Thus T_X is just the temperature at which the dynamics changes from flow-like (as proposed by the ideal version of MCT) to hopping like (generalized MCT). Thus the presented results are indeed coherent with a simple cross-over scenario predicted by theory.

I certainly do not want to force the authors to interpret their beautiful data by means of this theory, but they should at least think about in this direction and/or come up with a better discussion of the meaning of the data. The present version of the manuscript is lacking a convincing discussion and hence I invite the authors to improve this point.

Reply: We really appreciate the Referee for his/her positive comments and useful suggestions for us to improve the manuscript. We note that the context of the old version of our manuscript could be somehow inappropriate, which leads the Referee to think that we were trying to connect the changes of the dynamics at $T = 470$ K to the existence of the JG process. As mentioned above, the discussion about the JG β process (which is Arrhenius) is to highlight the fact that, even if our study is at temperatures well above the α and JG β crossover (only occurs at larger Q 's), we observe more Arrhenius-like behavior at larger Q 's. These discussions have been carefully revised in the new manuscript, and additional discussions connecting our observation of the length scale dependent dynamics to the well-documented microscopic structure of CKN have been provided.

Report of Reviewer #3

(From the Referee) Reviewer #3 (Remarks to the Author): This manuscript reports on a neutron scattering investigation of the relaxation processes in the fragile glass-former CKN in the normal and supercooled liquid state. The study covers about 4 decades in time between a fraction of ps and few ns, and a wide-angle Q -range between 0.8 and 2.8 \AA^{-1} .

The same sample was already studied with this approach as reported in a series of papers appeared in the 80s and 90s: this is carefully discussed in the manuscript. While the quality of the new set of data is undoubtedly superior than in those previous studies, by choosing an extensively studied sample the authors have decided to face here the challenge of extracting qualitatively new results.

The main point the authors make here is that there would exist a characteristic length of 2.6 \AA below which the structural relaxation would become more Arrhenius-like and would display a temperature independent heterogeneity. The basis of these claims is in Fig.3.

I observe that the fit results for beta (Fig.3a) and for tau (Fig.3b) look extremely well aligned as compared to the reported errorbar (one standard error, as reported by the authors). It would be important to provide more detail about the fitting procedure that has been used.

Have all parameters been left free? How have been the errorbars estimated? In fact, when the authors fit these (fit) results with a linear temperature dependence (for the beta) or with the VFT law (for the tau), they obtain very precise values of the parameters (D, T0 and $d(\beta)/dT$).

Reply: We really appreciate that the Referee noticed this issue about the error bars for τ , and f . We rechecked our data and our previous fitting procedures, and we found that the error bars were indeed not properly calculated. We used the standard nonlinear fitting procedure with the Levenberg-Marquardt algorithm in OriginLab to perform the fittings. In our fittings, all parameters are left free and we used the instrumental weight method (weight factor = $1/\sigma_i^2$ where σ_i is the error bar sizes of the data being fitted, see <https://www.originlab.com/doc/Origin-Help/Fit-with-Err-Weight>). We notice that the error bars of the intermediate scattering functions are very small, but that of the fitting parameters, f , τ , and $d(\beta)/dT$ are large; on the other hand, the error bars of τ and $d(\beta)/dT$ are large, but the error bars of the parameters obtained from the fit to τ and $d(\beta)/dT$ are not that large. This is because in our previous fittings the parameter's standard errors do not reflect the magnitude of weight. More explanations can be found here <https://www.originlab.com/doc/Quick-Help/Chi-Se-Remain>.

In the revised manuscript, we have reanalyzed **all** the data using the more appropriate settings ensuring that the parameter's standard errors reflect the magnitude of weight. The results are not significantly changed, but more reasonable error bars are given for the parameters. In the reanalyzed data, the error bars of f , τ , and $d(\beta)/dT$ are much smaller than before because of the high accuracy of the measured intermediate scattering functions with very small error bars. Since the error bars of f , τ , and $d(\beta)/dT$ get smaller, the parameters obtained from the fit to τ and $d(\beta)/dT$ with new error bars are not significantly changed compared with the previous ones without considering the magnitude of weight.

(From the Referee) They also obtain values higher than Tg for several T0 values corresponding to fits of tau(Q) data at low-Q's. How do the authors interpret that? Do they believe that the VFT model is still adequate to describe these data sets? and then what does the observation of a change of the fitted parameters (D, T0) of the VFT law really mean?

Reply: For the present work, VFT is just used as an empirical model and the fitted parameters D and T_0 provide quantitative comparison of the deviation from Arrhenius behavior at different Q 's. As for the question of whether VFT is the correct model of glass transition, it is not within the scope of our manuscript.

For the relatively large values of T_0 at low- Q 's, i.e., 0.8 \AA^{-1} and 1.08 \AA^{-1} , we also see that the values of D are smaller than the others. From the plot below we can see that the D and T_0 obtained from the macroscopic shear relaxation time (dash line) agree with the mean value of those at the lower Q 's, indicating that the macroscopic relaxation in CKN reflects the average of the processes at the different microscopic length scales. Moreover, we find that at $Q < 2.4 \text{ \AA}^{-1}$, the Q -dependence of D follows basically the shape of the static structure factor $S(Q)$, and that of T_0 follow the shape of $-S(Q)$, i.e., the opposite trend of $S(Q)$. This observation indicates that, not only the parameters that describe the relaxation process at a constant temperature, but also their temperature dependence, could be affected by $S(Q)$, further evidencing the strong structure-dynamics correlation at the microscopic length scales.

(From the Referee) More on the technical side: did they investigate possible correlations between D and T_0 ? How would the chi-squared of the fits of the VFT-law to the $\tau(Q)$

**data change if both parameters would be fixed to the values found at intermediate Q 's?
The same questions hold for the data for beta.**

Reply: We have tried to fit the data by fixing T_0 or D . We attach below the case at $Q = 2.60 \text{ \AA}^{-1}$ for example, where the blue lines are the VFT fit with both T_0 and D free fitting parameters, the Reduced Chi-square (see <https://www.originlab.com/doc/Quick-Help/Bad-ReducedChiSqr> for the definition) is 2.03. The red curve in the left figure is the best fit with T_0 fixed at 340 K which is obtained from the macroscopic shear relaxation time and close to the value found at intermediate Q 's, we can see the fit is bad, and the Reduced Chi-square becomes 58.58, much larger than fitting T_0 . The red curve in the right figure is the best fit with D fixed at 1.53 which is obtained from the macroscopic shear relaxation time and close to the value found at intermediate Q 's; we can see that also in this case the fit is unsatisfactory, and the Reduced Chi-square becomes 76.53. Not shown here, but as it can now be expected the fit is even worse (Reduced Chi-square = 124.89) when both T_0 and D are fixed.

For the case of β , β is the slope of the linear fit to the $\ln \tau$ vs T plot at $T \leq 470 \text{ K}$, and is used to quantify the changing rate of τ and thus the relaxation time distribution with temperature change, not expected to correlate with other parameters. We can see from the plot above that the Q -dependence of β also follows roughly a $-S(Q)$ -like shape (opposite trend of $S(Q)$) at $Q < 2.4 \text{ \AA}^{-1}$, further indicating that, the temperature dependence of β is also affected by the inter-particle correlation of the nitrate ions which originates the primary peak of $S(Q)$.

(From the Referee) Other points:

i) How do the authors relate the disappearance of the temperature dependence of the stretching parameter to the fact that the relaxation time becomes more Arrhenius-like? I understand that this is their observation, but do the authors have an explanation for that?

Reply: This could be correlated with the microscopic structure of the material. By employing neutron diffraction and reverse Monte Carlo modeling, Tengroth et al. have investigated the structure of CKN in detail [Phys. Rev. B 64, 224207 (2001)], which suggests that the structure factor of CKN as observed by neutron scattering is dominated by the planar triangular nitrate ions which seem to orientationally cluster in chainlike aggregates, and the cations take up space in between the aggregates of the nitrate triangles. We argue that the dynamical behavior at the length scales of $> 2.6 \text{ \AA}$ ($Q < 2.4 \text{ \AA}^{-1}$) reflects to a large extent the relaxations of the nitrate-nitrate ion correlations, while at the length scales of $< 2.6 \text{ \AA}$ ($Q > 2.4 \text{ \AA}^{-1}$) the dynamics are dominated by the nitrate-cation (mostly oxygen-cation) correlations where the local environment is relatively more stabilized against temperature change due to Coulomb interactions, leading to the disappearance of the temperature dependence of and the fact that the relaxation time becomes more Arrhenius-like.

We have added this discussion in the revised manuscript.

(From the Referee) ii) Fig.S2c shows that the shape of the $\tau(T)$ curve changes with Q . I would like the authors to discuss that more in detail.

In fact, the same trend is not clear on the original data plotted in Fig.S2b together with the errorbars. Moreover, a VFT law has of course a temperature-dependent activation energy. This implies that different portions of it, corresponding to different times, necessarily display a different curvature. Can the authors exclude this scenario?

Reply: The VFT law implies a changing activation energy with temperature, more specifically, increasingly smaller activation energy upon decreasing temperature. Instead of excluding this scenario, we would say it is a good scenario to understand our observation. The more “bent” temperature dependence of the relaxation time for the higher Q 's ($Q > 2.4 \text{ \AA}^{-1}$) indicates that the activation energy at the local length scales are less temperature dependent than at larger length scales, i.e., at higher temperatures the activation energy at local length scales is larger than at larger length scales, in contrast, at lower temperatures it is smaller. This could be associated with the fact that the local environment is relatively more stabilized against temperature change due to the Coulomb interactions, leading to a more Arrhenius-like behavior.

For the data plotted in Fig.S2(b) of the old manuscript, i.e., the average relaxation time $\langle \tau_{\text{slow}} \rangle$ as a function of $1000/T$, the same trend as τ_{slow} (the left panel in the figure below) can also be appreciated in the normalized $\langle \tau_{\text{slow}} \rangle$ without fitting, as shown in the right panel of the figure below. This reinforces the conclusion that the temperature dependence of the relaxation time is different at different Q -range. These two plots are also shown in Fig. 5 of the new manuscript.

(From the Referee) iii) It would be useful to show the results of Fig.2a in linear scale, maybe normalizing the $\tau(Q)$ data collected at the different temperatures. Would the comparison to $S(Q)/Q^2$ remain convincing even in this representation?

Reply: Yes, the comparison to $S(Q)/Q^2$ remains convincing even in this representation as shown below. The relaxation time at different temperatures is normalized at $Q = 1.86 \text{ \AA}^{-1}$, we can see that at $Q > 1.6 \text{ \AA}^{-1}$ the data collapse and agree with the shape of $S(Q)/Q^2$.

(From the Referee) iv) pag.2, second paragraph, 7th line - something is missing in that sentence.

Reply: This sentence has been modified in the revised manuscript.

On the basis of these considerations, I believe that while the main claims of the authors are potentially interesting, the effects they are behind are very small, and the presentation should be strengthened to clearly demonstrate that they are robust in fact against fitting uncertainties.

Reply: As can be seen from our responses to the comments of all three Referees, the main claims of the present work are robust against fitting uncertainties. The manuscript has been improved to strengthen these aspects. As far as the importance of this work is concerned, we underscore the

following additional points: 1) the observed Q -dependent temperature dependence of the α -relaxation time and the stretching exponent presents new phenomena for the microscopic relaxation dynamics, which is critical for understanding the dynamics evolution upon cooling towards the glass transition and is only enabled by the study across a wide temperature range and a large Q -range employing this newly developed neutron spin-echo spectrometer, 2) technically, it will present the first publication of the WASP spectrometer, the world's most powerful neutron spin-echo instrument, thus will be of broad interest.

REVIEWER COMMENTS

Reviewer #1 (Remarks to the Author):

The presented manuscript is the revised version of the work related to the neutron spin echo investigation of glass forming system $\text{Ca}_{0.4}\text{K}_{0.6}(\text{NO}_3)_{1.4}$ (CKN). Using the newly developed neutron spin echo instrument WASP, the authors focused their investigation mainly on a high temperature range of $T > 400\text{K}$ and relatively high values of Q . They discover substantial differences in the dynamics (Q and temperature behaviour) below and above Q value of the first diffraction peak. I think the work contains interesting and novel information for the research on glasses and amorphous systems. However, though the manuscript has been improved, it still contains a lot of hand waving arguments and in my opinion, needs further text revisions for reader's benefit. I have following comments and suggestions:

1. A very important role in the theoretical description of the glass transition and relaxation processes in highly viscous liquids plays Mode Coupling Theory (MCT). The real understanding of the results in the presented work without a proper introduction of mode coupling theory is in my opinion difficult. Therefore, MCT and its central predictions (two-steps relaxation processes; introduction of ergodicity parameter and critical temperature T_c) must be reflected in the text.
2. I am pleased that authors introduced the figure of the static structure factor into the main manuscript. However, I think it is not sufficient to understand the complex picture of interatomic/ionic correlations existing at different length scales. A graphic illustration of CKN structure with indicated correlation lengths would strongly benefit the readers.
3. I did not find anywhere in the text how the glass transition temperature of the samples has been verified. Were DSC measurements done? It could also be done (though not so accurately) using mean square displacement.
4. The authors use the terms "coherent" and "incoherent" in regard to neutron scattering signal without the proper definition what does it mean and the relation of these features to the results of the study. The authors should introduce these features correctly. Another alternative would be to simply describe neutron scattering signal in terms of pair-correlation functions and self-correlation function.
5. On the page 9 authors write: "the local environment is relatively more stabilized against temperature change due to the Coulomb interactions, leading to a more Arrhenius-like behaviour", having in mind correlations at $Q \geq 2.4 \text{ \AA}^{-1}$. However, looking at the static structure factor one can see that the temperature dependence of the second peak is very strong and shows no peak anymore already at $T \geq 420\text{K}$. This could be an indication that all intermolecular correlations decay very fast with temperature. In this case, I would expect that the signal will be caused mostly by the self-correlation part in the intermediate scattering function. Can this be the explanation of the changes in the temperature behaviour of β -parameters?

6. I understand that leaving all fitting parameters free would produce a better agreement with the experimental points. However, the Q dependences of the D and T₀ parameters on the figure 4d, which are in exact antiphase to each other, leave the impression that these parameters are interdependent. It would be good to see how Q dependence of T₀ looks like if D is fixed to the values established from the viscosity.

7. Since the work focuses on dynamic heterogeneities in CKN, it would be advisable to mention the observation of the dynamic heterogeneities in CKN at the pre-peak at $Q \sim 0.8 \text{ \AA}^{-1}$ originating from the string like local flow of ionic groups by F. Mezei et al for the complete picture.

8. On the page 6 the authors write: "possible relaxation processes and thus larger χ is observed, reflecting the more coherent structure of the liquid at this Q." What does "more coherent structure of the liquid" mean?

I recommend the manuscript revision with following resubmission.

Reviewer #2 (Remarks to the Author):

In the new version of the manuscript the authors have taken into account in a reasonable way my comments and also answered the questions I asked. They have replaced some of the figures by new ones and added several other figures in the SI. As a consequence the manuscript has become now much more accessible and interesting and hence I recommend it to be accepted.

Just a few minor things:

In the abstract: "in the equilibrium and highly viscous state". Formulated in this way one gets the impression that "equilibrium" and "highly viscous" are complementary and hence this is confusing. So I suggest to remove the "equilibrium and" so that things become clearer.

P. 2: "above or moderately below their melting temperature". This is not very precise since MCT makes no reference to the melting temperature. Why not refer to this T-regime as "moderately viscous state"?

P. 5: Here the authors refer to Fig. S2, a figure that they have added in the SI. In fact this figure is highly interesting since, without any fitting, one demonstrates the so-called alpha-scale universality, i.e. the fact that "all" observables show the same dependence on T. The authors might want to point this out to the reader.

P. 9: The authors write that the data in the Arrhenius plot is "bent". It might be more appropriate to say that the activation energy depends on T, since "bending" gives the impression that there is a kink.

Reviewer #3 (Remarks to the Author):

I find the present version of the manuscript considerably changed with respect to the previous one.

Based on a nice set of data collected using the new WASP spectrometer, the authors report that there would exist a characteristic length of 2.6 Å in the glass-former CKN below which the structural relaxation would become more Arrhenius-like and with a temperature independent distribution.

With respect to the previous version of the manuscript, the data

analysis results have been made more convincing thanks to a

more appropriate evaluation of error bars. The cross-over length-scale is here related no longer to the Johari-Goldstein relaxation as in the previous version of the manuscript but rather to the structure: above 2.6 Å the relaxation would be dominated by the inter-particle correlation of nitrate ions, while below 2.6 Å the temperature independence of the distribution of characteristic times and the more Arrhenius behaviour would be related to the "stabilization mechanism of the Coulomb interaction between nitrates and cations".

I am not sure of what this really means; I observe that the

high-Q region (above $2A^{-1}$) where this stabilization mechanism is claimed to be important is exactly the Q-range where the structure factor in Fig. 1 shows a clear temperature dependence. In other terms the dynamics is more Arrhenius and the distribution of relaxation time more temperature independent where the structure is more temperature dependent. I believe that this point should be addressed.

More in general, the description of the evolution of the slow

relaxation with Q and temperature in terms of structural motifs

makes the results necessarily system-dependent; in fact, the results obtained by the same group on ZnCl_2 (Ref.64) and recently published show a somewhat different scenario. It is therefore unclear to me at this stage how relevant are these results for the community interested in glass-formers and in the glass-transition problem.

I also find the new version of the manuscript more involved in terms of presentation. For example, the data reported in Fig.6 at the end of the manuscript are shown to demonstrate the very nice agreement of two sets of data obtained using two different spectrometers. This is certainly nice; but the fast relaxation appearing in those combined spectra is going to be discussed in a separate publication, and it is unclear to me why these data are reported here and, in particular, just before the conclusions.

On the basis of these considerations, I believe that while the quality of the data reported here is very high, the advancement in terms of understanding of the high-Q dynamics in glass-forming liquids is unclear.

Manuscript ID: NCOMMS-21-37605

Title: **"Q-dependent Collective Relaxation Dynamics of Glass-Forming Liquid Ca_{0.4}K_{0.6}(NO₃)_{1.4} Investigated by Wide-Angle Neutron Spin-Echo"** by P. Luo, et al.

Response letter to the comments of the Referees

*We thank the Referees for their valuable comments and suggestions. We reproduce below the original comments of the Referees marked in **bold**, followed by our point-by-point reply. We have also modified the text of the manuscript accordingly and hope the Referees find the present version suitable for publication in Nature Communications.*

Report of Reviewer #1

(From the referee) The presented manuscript is the revised version of the work related to the neutron spin echo investigation of glass forming system Ca 0.4K 0.6(NO₃)_{1.4} (CKN). Using the newly developed neutron spin echo instrument WASP, the authors focused their investigation mainly on a high temperature range of T>400K and relatively high values of Q. They discover substantial differences in the dynamics (Q and temperature behaviour) below and above Q value of the first diffraction peak. I think the work contains interesting and novel information for the research on glasses and amorphous systems. However, though the manuscript has been improved, it still contains a lot of hand waving arguments and in my opinion, needs further text revisions for reader's benefit. I have following comments and suggestions:

1. A very important role in the theoretical description of the glass transition and relaxation processes in highly viscous liquids plays Mode Coupling Theory (MCT). The real understanding of the results in the presented work without a proper introduction of mode coupling theory is in my opinion difficult. Therefore, MCT and its central predictions (two-

steps relaxation processes; introduction of ergodicity parameter and critical temperature T_c) must be reflected in the text.

Reply: These aspects have been introduced in the revised manuscript.

(From the referee) 2. I am pleased that authors introduced the figure of the static structure factor into the main manuscript. However, I think it is not sufficient to understand the complex picture of interatomic/ionic correlations existing at different length scales. A graphic illustration of CKN structure with indicated correlation lengths would strongly benefit the readers.

Reply: Such an illustration has been inserted in Figure 1.

(From the referee) 3. I did not find anywhere in the text how the glass transition temperature of the samples has been verified. Were DSC measurements done? It could also be done (though not so accurately) using mean square displacement.

Reply: We measured the glass transition for our sample using DSC, at a heating rate of 5 K min^{-1} . The onset glass transition temperature T_g is 336 K as shown in the following figure. It is consistent with previous reports. This figure has been added to Figure S1 into the Supplementary Materials and a reference made in the main manuscript.

(From the referee) 4. The authors use the terms “coherent” and “incoherent” in regard to neutron scattering signal without the proper definition what does it mean and the relation of these features to the results of the study. The authors should introduce these features

correctly. Another alternative would be to simply describe neutron scattering signal in terms of pair-correlation functions and self-correlation function.

Reply: We have clarified these terminologies in the revised manuscript as suggested by the referee.

(From the referee) 5. On the page 9 authors write: “the local environment is relatively more stabilized against temperature change due to the Coulomb interactions, leading to a more Arrhenius-like behaviour”, having in mind correlations at $Q \geq 2.4 \text{ \AA}^{-1}$. However, looking at the static structure factor one can see that the temperature dependence of the second peak is very strong and shows no peak anymore already at $T \geq 420 \text{ K}$. This could be an indication that all intermolecular correlations decay very fast with temperature. In this case, I would expect that the signal will be caused mostly by the self-correlation part in the intermediate scattering function. Can this be the explanation of the changes in the temperature behaviour of β -parameters?

Reply: We do not think that these dynamic features at $Q > 2.4 \text{ \AA}^{-1}$ are a result of the contribution from the self-correlation part in the intermediate scattering function (incoherent scattering). We have performed polarized diffraction measurements on the NGA-NSE spectrometer at NCNR for our CKN sample at $0.4 \text{ \AA}^{-1} \leq Q \leq 1.7 \text{ \AA}^{-1}$. Figure (a) below shows that the incoherent scattering intensity remains independent of Q and temperature, as expected. The total scattering intensity, which is the sum of incoherent and coherent scattering is known at higher Q 's, from the AMATERAS data, and the incoherent scattering intensity is not expected to change, which thus allows for the estimation of the incoherent contribution. Taking $T = 480 \text{ K}$ as an example, the static structure factor obtained from the measurements on AMATERAS at J-PARC is normalized to overlap with the total scattering obtained from polarized diffraction [see below Figure (b)]. The total scattering intensity at $Q = 2.4 \text{ \AA}^{-1}$ is thus 9.64, while the incoherent scattering intensity is 2 being the same as lower Q 's. Note that in NSE measurements, polarized neutrons are used, and the dynamic signal is given by $I_{\text{coh}} - 1/3I_{\text{inc}}$, therefore, the incoherent contribution at $Q = 2.4 \text{ \AA}^{-1}$ is $1/3 * 2 / (1/3 * 2 + 9.64) = 6.4\%$. As a comparison, we now look at $Q = 1 \text{ \AA}^{-1}$, where the coherent scattering intensity is just 2.92, and thus the incoherent contribution is $1/3 * 2 / (1/3 * 2 + 2.92) = 18.6\%$, which should be the largest value for CKN in the studied Q range of our spin echo experiments, from 0.8 \AA^{-1} to 2.82 \AA^{-1} . These numbers indicate that the

incoherent contribution is very small in CKN in our studied Q range, and the anomalous dynamic behavior observed for $Q > 2.4 \text{ \AA}^{-1}$ cannot be a result of incoherent scattering. If it were, it will be more pronounced around $Q = 1 \text{ \AA}^{-1}$ than at $Q > 2.4 \text{ \AA}^{-1}$. Furthermore, as shown in Fig. 6 of the manuscript, the agreement between the intermediate scattering functions obtained from neutron spin echo and time-of-flight experiments, which involves contribution from both coherent and incoherent scattering, indicate that incoherent contribution is negligible. Furthermore, it should be noted that although coherent scattering also has a single-particle contribution, strong temperature dependence of and non-Arrhenius behavior have often been observed at Q values lower than the main structure peak in multicomponent glass-forming systems like alloy liquids even when incoherent neutron scattering dominates [PRL 101, 037801 (2008); Appl. Phys. Lett. 95, 191907 (2009)]. Therefore, we conclude that our observed dynamic features are not a result of an increased contribution of self-correlation in the intermediate scattering function.

Instead, we consider the following to explain our experimental observation. In addition to the neutron diffraction data, a reverse Monte Carlo study on CKN by Tengroth et al. [Phys. Rev. B 64, 224207 (2001)] revealed that, while the decrease in the N-N coordination number at larger distances is the same as expected from thermal expansion, at local distances it is much larger. They attributed this to an inhomogeneous thermal expansion in the material; the nitrate-cation distances are stabilized by the Coulombic interactions and will not increase as much as nitrate-

nitrate ion distances upon heating. This difference is local (it extends not far beyond the nearest nitrate-nitrate ion distance) and is averaged out at larger distances, as expected. In this regard, the decreased scattering intensity above the structure factor main peak at $Q > 1.8 \text{ \AA}^{-1}$ upon increasing temperature is a result of the anomalous decrease of the N-N coordination number. Therefore, our observation of weaker temperature dependence of the dynamics at the length scales corresponding to the ionic oxygen-cation correlations can be explained by the stabilization mechanism of local environment due to Coulombic interactions, not contradicting the decreasing intensity of $S(Q)$ upon increasing temperature.

These discussions have been added to the revised manuscript and the Supplementary Information accordingly.

(From the referee) 6. I understand that leaving all fitting parameters free would produce a better agreement with the experimental points. However, the Q dependences of the D and T_0 parameters on the figure 4d, which are in exact antiphase to each other, leave the impression that these parameters are interdependent. It would be good to see how Q dependence of T_0 looks like if D is fixed to the values established from the viscosity.

Reply: When D is fixed to 1.53 as established from the viscosity, $T_0 \approx 340 \text{ K}$ is independent of Q , as shown in the figure below (blue squares in the lower panel) and is in agreement with that obtained from the viscosity. The quality of the fitting is however much worse in this case as indicated by the reduced Chi-square (blue squares in the upper panel). These results are not suggestive of overfitting or of an interdependence of D and T_0 , but indicate that the temperature dependence of the relaxation time is Q dependent. In the VFT equation, a larger value of the strength parameter D corresponds to a smaller deviation from Arrhenius behavior [see Böhmer, R., Ngai, K. L., Angell, C. A. & Plazek, D. J. Nonexponential relaxations in strong and fragile glass formers. *J. Chem. Phys.* 99, 4201–4209 (1993)]. On the other hand, Arrhenius behavior is the limit of VFT when T_0 approaches 0, therefore, for the same sample, a smaller T_0 means more Arrhenius-like behavior. In this regard, an interdependence of D and T_0 is expected for varying degree of deviation from Arrhenius behavior. Therefore, instead of an overfit, the antiphase Q dependence of D and T_0 is indeed an indication of varied degree of deviation from Arrhenius behavior, consistent with the raw data shown in Fig. 5 of the manuscript. If the data were overfitted, the reduced Chi-square should not change much when one parameter is fixed.

(From the referee) 7. Since the work focuses on dynamic heterogeneities in CKN, it would be advisable to mention the observation of the dynamic heterogeneities in CKN at the pre-peak at $Q \sim 0.8 \text{ \AA}^{-1}$ originating from the string like local flow of ionic groups by F. Mezei et al for the complete picture.

Reply: We thank the Referee for this suggestion. Actually, the string-like flow proposed by Mezei et al. [*Phys. B Condens. Matter* **276–278**, 437–439 (2000)] is consistent with the later observation of chainlike aggregates of nitrate ions by Tengroth et al. [*Phys. Rev. B* **64**, 224207 (2001)]. This has been mentioned in the revised manuscript.

(From the referee) 8. On the page 6 the authors write:” possible relaxation processes and thus larger is observed, reflecting the more coherent structure of the liquid at this Q.” What does “more coherent structure of the liquid” mean?

Reply: “More coherent structure” means more pronounced structure correlations at this length scale. We have changed “more coherent structure” in this sentence to “more pronounced structure correlation” in the revised manuscript.

I recommend the manuscript revision with following resubmission.

Reply: We believe that all the concerns of the Referee have been addressed and hope that the Referee is satisfied with the above reply.

Report of Reviewer #2

(From the referee) In the new version of the manuscript the authors have taken into account in a reasonable way my comments and also answered the questions I asked. They have replaced some of the figures by new ones and added several other figures in the SI. As a consequence the manuscript has become now much more accessible and interesting and hence I recommend it to be accepted.

Just a few minor things:

In the abstract: "in the equilibrium and highly viscous state". Formulated in this way one gets the impression that "equilibrium" and "highly viscous" are complementary and hence this is confusing. So I suggest to remove the "equilibrium and" so that things become clearer.

Reply: We thank the Referee pointing this out, “equilibrium and” has been removed.

(From the referee) P. 2: "above or moderately below their melting temperature". This is not very precise since MCT makes no reference to the melting temperature. Why not refer to this T-regime as "moderately viscous state"?

Reply: Here with the expression of “below their melting temperature” we refer to the temperature range of supercooled liquid state in real glasses. But the Referee is right, it would be better to say “moderately viscous liquid state”. We have thus changed it in the revised manuscript.

(From the referee) P. 5: Here the authors refer to Fig. S2, a figure that they have added in the SI. In fact this figure is highly interesting since, without any fitting, one demonstrates

the so-called alpha-scale universality, i.e. the fact that "all" observable show the same dependence on T. The authors might want to point this out to the reader.

Reply: In the Q range of $1.8 \text{ \AA}^{-1} < Q < 2.4 \text{ \AA}^{-1}$ the data agree well in this plot, which is consistent with the very small change of D and T_0 from VFT fit as can be seen in Fig. 4(d). But for $Q > 2.4 \text{ \AA}^{-1}$, the dispersion of normalized data is well beyond the error bar, although basically the data at each temperature follow the shape of $S(Q)/Q^2$ at $Q > 1.8 \text{ \AA}^{-1}$. Also, Fig. 5 in the manuscript has the relaxation time normalized to a constant temperature indicates that the temperature dependence at $Q > 2.4 \text{ \AA}^{-1}$ is different from that around the structure factor main peak. These results indicate that the alpha-scale universality may exist in a specific range of length scales, but the modulation of the local environment is not trivial. On the other hand, previous studies [PRL 109, 115705 (2012); J. Chem. Phys. 140, 144906 (2014)] employing nuclear resonance X-ray scattering have shown that at $\sim 10^{-7}$ s, i.e., just one order of magnitude slower than the slowest time in our study, the relaxation time at the Q range of the minimum above the structure factor main peak changes from VFT behavior to Arrhenius behavior associated with JG-relaxation. Therefore, we remain cautious about making a conclusion about the alpha-scale universality by adding such a sentence in the caption of Fig. S3, "The good agreement between the normalized relaxation time at $1.8 \text{ \AA}^{-1} < Q < 2.4 \text{ \AA}^{-1}$ is consistent with the very small change of D and T_0 from VFT fit as shown in Fig. 4(d), suggesting the similar temperature dependence of the relaxation time in this Q range."

(From the referee) P. 9: The authors write that the data in the Arrhenius plot is "bent". It might be more appropriate to say that the activation energy depends on T, since "bending" gives the impression that there is a kink.

Reply: We have removed "bent" and reworded accordingly.

Report of Reviewer #3

(From the referee) I find the present version of the manuscript considerably changed with respect to the previous one.

Based on a nice set of data collected using the new WASP spectrometer, the authors report that there would exist a characteristic length of 2.6 Å in the glass-former CKN below which the structural relaxation would become more Arrhenius-like and with a temperature independent distribution.

With respect to the previous version of the manuscript, the data analysis results have been made more convincing thanks to a more appropriate evaluation of errorbars. The crossover length-scale is here related no longer to the Johari-Goldstein relaxation as in the previous version of the manuscript but rather to the structure: above 2.6 Å the relaxation would be dominated by the inter-particle correlation of nitrate ions, while below 2.6 Å the temperature independence of the distribution of characteristic times and the more Arrhenius behaviour would be related to the "stabilization mechanism of the Coulomb interaction between nitrates and cations".

Reply: We believe that the Referees have misunderstood our expressions in the previous version of our manuscript. In the first submitted version of the manuscript, we mentioned that our experiments are above the alpha - Johari-Goldstein beta bifurcation temperature and we did not relate our observations to the Johari-Goldstein relaxation. This was also explained in our reply to the Referees in the last round.

In the present revised manuscript, we are more careful with our wording. However, we do not exclude the possibility that our observed more Arrhenius-like relaxation at local length scales is related with the Johari-Goldstein relaxation. As a matter of fact, even today the origin of the Johari-Goldstein relaxation is still debated. Studies employing nuclear resonance X-ray scattering (NRXS) revealed that the crossover of alpha-beta relaxation (transition of non-Arrhenius to Arrhenius behavior) only occurs at the local length scales corresponding to the minimum above the structure factor main peak [PRL 109, 115705 (2012); J. Chem. Phys. 140, 144906 (2014)]. Our observation of a stronger Arrhenius-like behavior and temperature independence of τ , at the local length scales above the crossover temperature of alpha-beta relaxation, leads us to ask such questions: Is the Johari-Goldstein relaxation fully covered by the structural relaxation above the crossover temperature or does it have distinct features from the structural relaxation even above the crossover temperature? What is the role of the local

environment in the appearance of Johari-Goldstein relaxation? These questions need to be answered for an in-depth understanding of the Johari-Goldstein relaxation.

(From the referee) I am not sure of what this really means; I observe that the high-Q region (above 2 \AA^{-1}) where the this stabilization mechanism is claimed to be important is exactly the Q-range where the structure factor in Fig.1 shows a clear temperature dependence. In other terms the dynamics is more Arrhenius and the distribution of relaxation time more temperature independent where the structure is more temperature dependent. I believe that this point should be addressed.

Reply: The previous neutron diffraction and reverse Monte Carlo study on CKN by Tengroth et al. [Phys. Rev. B 64, 224207 (2001)] revealed that, while the decrease in the N-N coordination number at larger distances is the same as expected from thermal expansion, at local distances it is much larger. They attributed this to an inhomogeneous thermal expansion in the material; the nitrate-cation distances are stabilized by the Coulombic interactions and will not increase as much as nitrate-nitrate ion distances upon heating. This difference is local (it extends not far beyond the nearest nitrate-nitrate ion distance) and is averaged out at larger distances, as expected. In this regard, the decreased scattering intensity above the structure factor main peak at $Q > 1.8 \text{ \AA}^{-1}$ upon increasing temperature is a result of the anomalous decrease of the N-N coordination number. Therefore, our observation of weaker temperature dependence of the dynamics at the length scales corresponding to the ionic oxygen-cation correlations can be explained by the stabilization mechanism of local environment due to the Coulombic interactions, not contradicting the decreasing intensity of $S(Q)$ upon increasing temperature.

An explanation around these concerns has been added to the revised manuscript.

(From the referee) More in general, the description of the evolution of the slow relaxation with Q and temperature in terms of structural motifs makes the results necessarily system-dependent; in fact, the results obtained by the same group on ZnCl_2 (Ref.64) and recently published show a somewhat different scenario. It is therefore unclear to me at this stage how relevant are these results for the community interested in glass-formers and in the glass-transition problem.

Reply: For the previous NSE measurements on ZnCl_2 , an anomalous increase of the exponent was observed for the Q 's corresponding to the structure factor main peak and pre-peak upon decreasing the temperature towards the melting point. This is related to the formation of 3D network structure by the tetrahedral motifs. ZnCl_2 is a highly unusual molten salt – as peculiar as water because of its tetrahedra network structure. It is extremely weakly ionic, with an electronegativity difference of 1.5. It has an extremely low melting point (583 K) compared to other salts. It also has a very low fragility of 30 – closer to that observed in strong liquids. On the other hand, CKN is a much better representative of fragile liquids. The present study for CKN covers the high Q range that has not been studied before. Whether similar phenomena as that observed in CKN at higher Q 's in the supercooled liquid region, exists in ZnCl_2 , remains an open question. A comparison between the two would be very interesting.

Before the present work, the relaxation dynamics at high Q 's (beyond the structure factor main peak), had never been studied in such detail. This is now possible owing to the new capabilities offered by the WASP instrument. For a prototypical glass-forming system like CKN, an in-depth understanding of the relaxation dynamics is important. As also mentioned in the manuscript, whether the effects observed in CKN are universal to other glass forming systems with distinct nature of interatomic/molecular interactions, remains an open question (although our intuition is yes), and we expect that knowledge of the relaxation properties at local length scales in different glass forming systems may help understand the intrinsic origin of their distinct dynamic behaviors, like the different appearance of Johari-Goldstein relaxation and fragility.

(From the referee) I also find the new version of the manuscript more involved in terms of presentation. For example, the data reported in Fig.6 at the end of the manuscript are shown to demonstrate the very nice agreement of two sets of data obtained using two different spectrometers. This is certainly nice; but the fast relaxation appearing in those combined spectra is going to be discussed in a separate publication, and it is unclear to me why these data are reported here and, in particular, just before the conclusions.

Reply: The present paper aims mainly to report the contrasting long time scale relaxation dynamics at different length scales, which is enabled only by the newly commissioned WASP instrument. The nice agreement of the WASP data for the prototypical glass former CKN and that obtained from a time-of-flight instrument is technically very important, since in previous

experiments including those on other glass forming systems, the separate data sets have had to be artificially adjusted to force a match. By being able to cover the full range of time scales, a single dataset can be fitted by Equation (1) with the parameters obtained from the WASP data only, confirming that our results from the present work are not an artifact of data fitting. Therefore, we think it is necessary to present the Fig. 6 in this paper. For the fast relaxation we have performed a more systematic study and have other findings which are best separated from this manuscript. The focus here is on the long time structural relaxations.

(From the referee) On the basis of these considerations, I believe that while the quality of the data reported here is very high, the advancement in terms of understanding of the high-Q dynamics in glass-forming liquids is unclear.

Reply: We hope the above point-by-point replies address all the Referee's concerns and that they will recommend our work for publication.

REVIEWERS' COMMENTS

Reviewer #1 (Remarks to the Author):

Review report of the manuscript NCOMMS-21-37605 "Q-dependent Collective Relaxation Dynamics of Glass-Forming Liquid $\text{Ca}_{0.4}\text{K}_{0.6}(\text{NO}_3)_{1.4}$ Investigated by Wide-Angle Neutron Spin-Echo" by YZ and collaborators

The presented manuscript describes the results of the neutron scattering study of the ionic glass former $\text{Ca}_{0.4}\text{K}_{0.6}(\text{NO}_3)_{1.4}$ (CKN). CKN is well known glass former materials which consists of ions of different sizes and is used quite often as a closest approximation of Lennard Jones system. Previously, strong non-Arrhenius behavior for CKN viscosity in this glass former has been found by A. Angell. Two steps of relaxation processes at nano and picosecond time scale have been found by F. Mezei and coworkers. In the present work the authors have concentrated on the slow structural relaxation mostly above the main peak in the structure factor at 1.8 \AA^{-1} , which was not investigated before and is the major achievement of this study. Thus, they have found striking differences in microscopic dynamics at local and intermediate length scales using moderate fitting and no modelling.

I think that the results of the study have a sufficient novelty and are important for the science of disordered and amorphous materials. After two revisions the quality of the manuscript has been substantially improved. The current state of the art is now much better addressed and provides a good basis for the understanding of the presented study. The recently introduced figure 1 with measured static structure factor and the cartoon, illustrating different length scale in CKN, improves considerably the reading. The quality of the scientific data has been also improved. Thus, the authors included the data illustrating the determination of the glass transition temperature and the fraction of the incoherent scattering, addressing two important issues from previous review. The scientific argumentation is now more precise and justified. Based on results and the quality of the presented manuscript I recommend it now for publication with minor modifications which will not need an additional review. In particular:

1. On the Figure 1 the caption sites the reason for the second peak in the structure factor at 2.85 \AA^{-1} as ionic oxygen-cation correlations. Looking at the picture insert I have rather an impression that correlations between an oxygen and K^+ ion lead to this peak. Is this what authors wish to express?
2. On the page 6 and 7 last and first paragraph correspondingly, it is written: "narrower distribution of possible relaxation processes and thus larger τ is observed, reflecting the more pronounced structure correlation of the liquid at this Q. However, at Q's outside this length scale, the structural correlation becomes weaker...."- are the authors sure they want to say singular "correlation" instead of plural "correlations"?
3. It is not obligatory, but I would recommend giving a critical look to the discussion on the pages 10 and 11, because some statements are repeated.

I congratulate the authors on this hard and challenging work and on interesting results presented.

Reviewer #2 (Remarks to the Author):

The authors have taken into account all my previous comments and therefore I recommend the manuscript to be accepted.

Reviewer #3 (Remarks to the Author):

I see this manuscript for the third time. The question of how representative are the reported results for viscous liquids remains unanswered to me. However, I also think that the authors addressed quite well the various issues raised by the referees, and I am not against the publication of this manuscript in Nature Communications.

Manuscript ID: NCOMMS-21-37605B

Title: **"Q-dependent Collective Relaxation Dynamics of Glass-Forming Liquid Ca_{0.4}K_{0.6}(NO₃)_{1.4} Investigated by Wide-Angle Neutron Spin-Echo"** by P. Luo, et al.

Response letter to the comments of the Referees

*We reproduce below the Referees' comments marked in **bold**, followed by our point-by-point reply. We have also modified the text of the manuscript accordingly. We really appreciate the valuable comments and suggestions from the Referees in the entire review rounds, which are critical for improving our manuscript.*

Report of Reviewer #1

(From the referee) The presented manuscript describes the results of the neutron scattering study of the ionic glass former Ca_{0.4}K_{0.6}(NO₃)_{1.4} (CKN). CKN is well known glass former materials which consists of ions of different sizes and is used quite often as a closest approximation of Lennard Jones system. Previously, strong non-Arrhenius behavior for CKN viscosity in this glass former has been found by A. Angell. Two steps of relaxation processes at nano and picosecond time scale have been found by F. Mezei and coworkers. In the present work the authors have concentrated on the slow structural relaxation mostly above the main peak in the structure factor at 1.8 \AA^{-1} , which was not investigated before and is the major achievement of this study. Thus, they have found striking differences in microscopic dynamics at local and intermediate length scales using moderate fitting and no modelling.

I think that the results of the study have a sufficient novelty and are important for the science of disordered and amorphous materials. After two revisions the quality of the manuscript has been substantially improved. The current state of the art is now much

better addressed and provides a good basis for the understanding of the presented study. The recently introduced figure 1 with measured static structure factor and the cartoon, illustrating different length scale in CKN, improves considerably the reading. The quality of the scientific data has been also improved. Thus, the authors included the data illustrating the determination of the glass transition temperature and the fraction of the incoherent scattering, addressing two important issues from previous review. The scientific argumentation is now more precise and justified. Based on results and the quality of the presented manuscript I recommend it now for publication with minor modifications which will not need an

additional review. In particular:

1. On the Figure 1 the caption sites the reason for the second peak in the structure factor at 2.85 \AA^{-1} as ionic oxygen-cation correlations. Looking at the picture insert I have rather an impression that correlations between an oxygen and K^+ ion lead to this peak. Is this what authors wish to express?

Reply: No, the second peak in the structure factor at 2.85 \AA^{-1} should arise from both oxygen-potassium and oxygen-sodium correlations. We have modified the insert picture a little bit to make this clearer.

2. On the page 6 and 7 last and first paragraph correspondingly, it is written: “narrower distribution of possible relaxation processes and thus larger \square is observed, reflecting the more pronounced structure correlation of the liquid at this Q . However, at Q 's outside this length scale, the structural correlation becomes weaker...”- are the authors sure they want to say singular “correlation” instead of plural “correlations”?

Reply: We agree that plural “correlations” is better, and we have changed this in the revised manuscript.

3. It is not obligatory, but I would recommend giving a critical look to the discussion on the pages 10 and 11, because some statements are repeated.

Reply: This statement in page 10 looks repeated and has been removed: “Our results reveal the characteristic spatial length of $2\pi/Q \approx 2.6 \text{ \AA}$ ($Q = 2.4 \text{ \AA}^{-1}$) below which the dynamics exhibit distinct features with less variations of the activation energy (more Arrhenius-like behavior of

τ_{slow}) and invariant dynamic heterogeneity upon temperature change, in stark contrast to the structure modulated relaxation dynamics above 2.6 \AA ($Q < 2.4 \text{ \AA}^{-1}$).”

I congratulate the authors on this hard and challenging work and on interesting results presented.

Report of Reviewer #2

(From the referee) The authors have taken into account all my previous comments and therefore I recommend the manuscript to be accepted.

Report of Reviewer #3

(From the referee) I see this manuscript for the third time. The question of how representative are the reported results for viscous liquids remains unanswered to me. However, I also think that the authors addressed quite well the various issues raised by the referees, and I am not against the publication of this manuscript in Nature Communications.